# Evaluating LLMs at Detecting Errors in LLM Responses

**Ryo Kamoi**[1]**, Sarkar Snigdha Sarathi Das**[1]**, Renze Lou**[1]**, Jihyun Janice Ahn**[1]
**Yilun Zhao**[2]**, Xiaoxin Lu**[1]**, Nan Zhang**[1]**, Yusen Zhang**[1]**, Ranran Haoran Zhang**[1]
**Sujeeth Reddy Vummanthala**[1]**, Salika Dave**[1]**, Shaobo Qin**[3]
**Arman Cohan**[2,4]**, Wenpeng Yin**[1]**, Rui Zhang**[1]
[1]Penn State University, [2]Yale University, [3]Stony Brook University, [4]Allen Institute for AI
{ryokamoi, rmz5227}@psu.edu

## Abstract

With Large Language Models (LLMs) being widely used across various tasks, detecting errors in their responses is increasingly crucial. However, little research has been conducted on error detection of LLM responses. Collecting error annotations on LLM responses is challenging due to the subjective nature of many NLP tasks, and thus previous research focuses on tasks of little practical value (e.g., word sorting) or limited error types (e.g., faithfulness in summarization). This work introduces *ReaLMistake*, the first error detection benchmark consisting of objective, realistic, and diverse errors made by LLMs. *ReaLMistake* contains three challenging and meaningful tasks that introduce objectively assessable errors in four categories (reasoning correctness, instruction-following, context-faithfulness, and parameterized knowledge), eliciting naturally observed and diverse errors in responses of GPT-4 and Llama 2 70B annotated by experts. We use *ReaLMistake* to evaluate error detectors based on 12 LLMs. Our findings show: 1) Top LLMs like GPT-4 and Claude 3 detect errors made by LLMs at very low recall, and all LLM-based error detectors perform much worse than humans. 2) Explanations by LLM-based error detectors lack reliability. 3) LLM-based error detection is sensitive to small changes in prompts but remains challenging to improve. 4) Popular approaches to improving LLMs, including self-consistency and majority vote, do not improve the error detection performance. Our benchmark and code are provided at https://github.com/psunlpgroup/ReaLMistake.

## 1 Introduction

As LLMs have been increasingly used in real-world applications, it is critical to develop methods for automatically detecting errors in responses from LLMs (Bommasani et al., 2021;

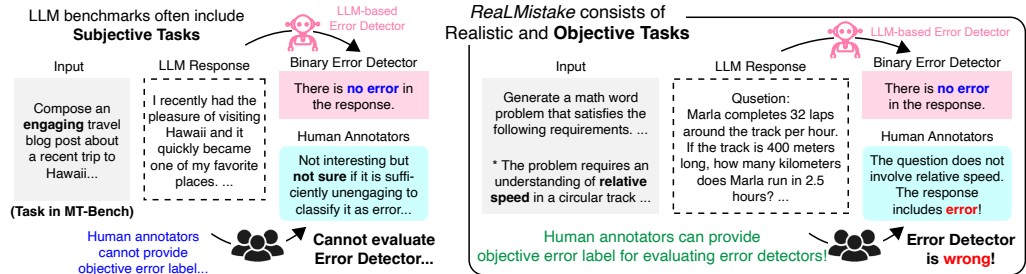

Figure 1: Left: Tasks in existing LLM evaluation benchmarks are often subjective and not suitable for collecting errors made by LLMs for the purpose of evaluating binary error detection methods. Right: We introduce the *ReaLMistake* benchmark with realistic, objective, and diverse errors made by LLMs for evaluating error detection.

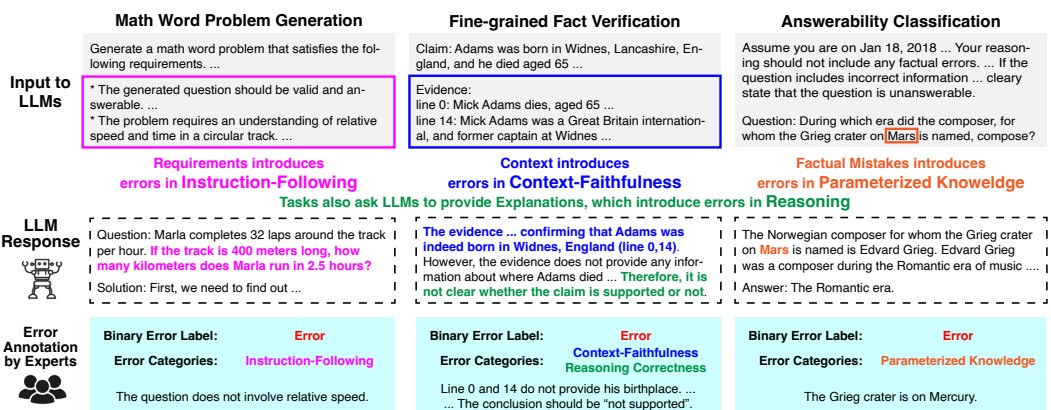

Figure 2: Examples of three tasks in *ReaLMistake* with four error categories. Each instance includes a binary error label, error categories, and annotator's explanations about errors on a response from GPT-4-0613 or Llama 2 70B. Appendix I provides full details.

| Benchmark | Task | Response Models | # Data | Label | No Subjective Criteria | Error Categories | | | |
|---|---|---|---|---|---|---|---|---|---|
| | | | | | | Reasoning Correctness | Instruction-Following | Context-Faithfulness | Parameterized Knowledge |
| MT-Bench, PandaLM, LLMEval^2 | Multiple NLP Tasks | Multiple LLMs | 480 - 2,553 | Ranking (Pairwise) | | ✓ | ✓ | ✓ | ✓ |
| WikiBio GPT-3 (Manakul et al., 2023) | Wikipedia Generation | GPT-3 | 238 | 3 options (Pointwise) | | | | | ✓ |
| SummEdits (Laban et al., 2023) | Summarization | GPT-3.5 Turbo | 6,348 | Binary (Pointwise) | | | | ✓ | |
| BIG-Bench Mistake (Tyen et al., 2024) | Logical Tasks (e.g., word sorting) | PaLM 2 | 2,778 | Binary (Pointwise) | ✓ | ✓ | ✓ | | |
| *ReaLMistake* (Ours) | 3 NLG Tasks in Different Domains | GPT-4 (0613), Llama 2 70B | 900 | Binary (Pointwise) | ✓ | ✓ | ✓ | ✓ | ✓ |

Table 1: Comparison between *ReaLMistake* and prior benchmarks for evaluating error detectors or evaluators for LLM responses. There is a deficiency in benchmarks for evaluating pointwise evaluation, although there are diverse benchmarks for evaluating pairwise evaluation (e.g., Zheng et al., 2023; Dubois et al., 2023; Zhang et al., 2023). *ReaLMistake* is the first benchmark that includes objective and realistic errors in LLM responses in diverse categories.

Srivastava et al., 2022). However, there is a deficiency in an analysis specifically targeting error detection (Huang et al., 2024; Tyen et al., 2024).

An obstacle in studying error detection is the lack of benchmarks that include binary error annotations (i.e., whether the response contains errors or not) on objective, realistic, and diverse errors made by LLMs.[1] An NLP task should satisfy the following criteria to be suitable for collecting errors made by LLMs for the purpose of benchmarking error detection. First, **to provide objective error labels, tasks should not involve subjectivity or ambiguity**. As illustrated in Figure 1, subjective tasks on which humans cannot provide reliable error annotation are not suitable for benchmarking error detection methods. For example, in text summarization, humans cannot objectively annotate binary error labels because the evaluation of "whether the generated summary properly includes important content in the source" (the "relevance" criterion) is subjective (Laban et al., 2023). It is difficult to evaluate error detectors if humans cannot determine whether LLM responses include errors. As shown in Table 1, tasks in benchmarks for ranking-based LLM evaluators (Zheng et al., 2023; Wang et al., 2023a; 2024) and criterion-specific error detectors (e.g., factuality evaluators) (Laban et al., 2023; Manakul et al., 2023) often involve subjectivity (Zeng et al., 2024). Second,

---

[1]In this paper, "objective errors" represent errors in LLM responses that do not involve ambiguity or subjectivity in the detection process and can be detected in a binary manner with a high inter-annotator agreement.

error detection benchmarks should be built on tasks that make LLMs introduce **errors similar to errors in real-world applications**. Some prior studies use tasks such as word sorting (Tyen et al., 2024) and chess (Wu et al., 2024), which are objective but have different properties from practical uses of LLMs. Third, to provide a comprehensive testbed, the benchmarks need to include a variety of tasks to provide **errors in diverse categories**. Previous research in detecting mistakes in LLM responses focuses on errors of limited types (Laban et al., 2023; Manakul et al., 2023; Lightman et al., 2024), as shown in Table 1. Finally, tasks should be **challenging for strong LLMs** to make them introduce errors.

To create tasks that satisfy these requirements, we propose an approach to design tasks so that they make LLMs introduce errors detected by objective, realistic, and diverse evaluation criteria. First, we identify four evaluation criteria (*Reasoning Correctness*, *Instruction-Following*, *Context-Faithfulness*, and *Parameterized Knowledge*) which can be objectively evaluated by humans and cover diverse errors in LLM responses. Subsequently, we create three tasks (*Math Word Problem Generation*, *Fine-grained Fact Verification*, and *Answerability Classification*) with the intention of making LLMs introduce errors detected by the four evaluation criteria, eliminating subjectivity from the error annotation process. This process ensures that the created tasks have desirable properties for collecting mistakes in LLM responses for error detection benchmarks. Using these three tasks, **we introduce *ReaLMistake*, the first benchmark for evaluating error detection methods for LLM responses, consisting of objective, realistic, and diverse errors made by LLMs**. As shown in Figure 2, our benchmark includes error annotations (binary error label, error categories, and human explanation about errors) on responses from GPT-4 (OpenAI, 2023) and Llama 2 70B (Touvron et al., 2023) on the three tasks. The annotation process requires careful checking of the entire LLM responses, and 14 expert annotators spent 90 hours to provide high-quality annotations.

By using *ReaLMistake*, we analyze the error detection performance of 12 LLMs (7 open-source and 5 closed-source models). Our findings show: (a) In the experiments on LLM-based error detectors with zero-shot prompting (Kojima et al., 2022), **top LLMs like GPT-4 and Claude 3 Opus detect errors made by LLMs at very low recall, and all models perform much worse than humans on the error detection task in *ReaLMistake***. (b) The performances of the error detectors on the three tasks in *ReaLMistake* have different trends, indicating that *ReaLMistake* provides error detection tasks with diverse properties. (c) Our manual analysis shows that explanations provided by the LLM-based error detectors are unreliable, and open-source LLMs often provide wrong reasoning even when the binary predictions are correct. (d) Our analysis of four different types of prompts for LLM-based error detection shows that error detection performance is sensitive to small differences in prompts. (e) Finally, to improve error detectors, we evaluate three methods motivated by popular approaches: self-consistency (Wang et al., 2023b), using multiple LLMs (Cohen et al., 2023; Chan et al., 2024), and providing evaluation steps in prompts (Liu et al., 2023a). However, we do not observe improvement in the error detection performance. Our analysis indicates that *ReaLMistake* provides challenging and diverse error detection tasks, and further research is needed to improve LLM-based error detectors for LLM responses.

## 2   Related Work

**Ranking-based Evaluation.**   Evaluation of LLMs has been mainly studied in ranking-based (Chen et al., 2023c; Dubois et al., 2023; Zheng et al., 2023; Wang et al., 2023a; 2024; Zeng et al., 2024) or Likert scale (Chiang & Lee, 2023) since recent LLMs often do not make obvious mistakes and many NLP tasks involve subjectivity. These evaluation approaches are useful for system-level comparison of LLMs but not suitable for detecting erroneous responses. Binary error detection is motivated by a real-world situation in which we want to automatically detect errors in LLM responses to reject or improve bad outputs.

**Self-Correction.**   Error detection is an important task in its own right, but it has been mainly studied in the context of improving LLM responses using feedback from LLMs (Bai et al., 2022; Madaan et al., 2023; Shinn et al., 2023; Kim et al., 2023; Chen et al., 2023a; Gou et al., 2024; Pan et al., 2024; Kamoi et al., 2024). These studies often evaluate the performance

| Response Model | Task | # Data | Average # tokens | | Errors in Responses from GPT-4 or Llama 2 70B [%] | | | | |
| --- | --- | --- | --- | --- | --- | --- | --- | --- | --- |
| | | | Input | LLM Response | Reasoning Correctness | Instruction- Following | Context- Faithfulness | Parameterized Knowledge | Total Error |
| GPT-4 0613 | Math Word Problem Generation | 140 | 252 | 151 | 25.0 | 57.1 | – | – | 62.1 |
| | Fine-grained Fact Verification | 140 | 523 | 83 | 25.7 | 5.7 | 45.0 | – | 62.9 |
| | Answerability Classification | 140 | 119 | 75 | 22.1 | – | 8.6 | 40.7 | 62.1 |
| Llama 2 70B | Math Word Problem Generation | 160 | 235 | 163 | 51.2 | 67.5 | – | – | 80.0 |
| | Fine-grained Fact Verification | 160 | 511 | 168 | 56.9 | 44.4 | 45.6 | – | 80.6 |
| | Answerability Classification | 160 | 119 | 96 | 48.1 | – | – | 48.1 | 81.2 |

Table 2: Statistics of our *ReaLMistake* benchmark.

of the refined outputs on downstream tasks and an analysis of the binary error detection step is lacking.

# 3 *ReaLMistake*

The *ReaLMistake* benchmark includes 900 instances consisting of error annotations by experts (binary error label: `error` or `no_error`, error categories, and human explanation) on responses from GPT-4-0613 (OpenAI, 2023) and Llama 2 70B (Touvron et al., 2023) on three tasks designed for providing objective error labels in diverse error categories, as shown in Table 2. Figure 2 provides examples from the three tasks in our benchmark.

## 3.1 How to Collect Objective, Diverse, and Realistic Errors in LLM Responses?

The main challenge in creating evaluation benchmarks for error detection is in the process of collecting errors in LLM responses. There are various requirements for tasks used to collect errors in LLM responses for the purpose of evaluating error detection. First, human annotators should be able to provide **objective** error labels for LLM responses on the tasks to create a reliable testbed of error detection, as shown in Figure 1. In addition, the tasks should introduce **diverse** types of errors observed in **real-world applications of LLMs**. Finally, the tasks should be **difficult even for recent LLMs** to make LLMs introduce mistakes. However, tasks should not be too difficult for humans to allow annotations and detailed analysis of error detection. We propose a strategy to construct tasks that satisfy these requirements.

**(1) Create Tasks that Make LLMs Introduce Objective, Realistic, and Diverse Errors.** Evaluation of NLP tasks often involves subjectivity and even humans cannot provide objective binary error labels. For example, the text summarization task involves subjectivity in the "relevance" criterion, which evaluates "whether important content in the source is properly selected" (Laban et al., 2023). To put it the other way around, humans can provide reliable error labels if tasks can be evaluated by objective criteria. To create tasks that make LLMs introduce objective, realistic, and diverse errors, **we propose a bottom-up approach to design tasks so that they can be evaluated using diverse and objective evaluation criteria**. We identify the following four criteria, which comprehensively cover objective criteria about errors in LLM responses that are caused by components of the inference process of LLMs: instructions and context in inputs, reasoning by LLMs, and knowledge in parameters. By creating tasks so that they can be evaluated by the following criteria, LLM responses to the tasks include realistic and diverse errors that can be objectively detected by humans.

- **Reasoning Correctness**: Reasoning in responses (e.g., chain-of-thought (Wei et al., 2022b)) should be logically valid. This criterion is objective because logical correctness is well-defined. Prior studies also annotate logical mistakes in reasoning by LLMs in a binary manner (Ling et al., 2023; Miao et al., 2023; Tyen et al., 2024; Lightman et al., 2024).

- **Instruction-Following**: Responses should follow the requirements specified in inputs (Zhou et al., 2023a; Zeng et al., 2024; Qin et al., 2024).[2] Instruction-following is an objective criterion if the requirements are objective (Zhou et al., 2023a).

---

[2]Although the term "instruction-following" is used in various scopes (Wei et al., 2022a; Ouyang et al., 2022), we use a definition that focuses on the requirements explicitly mentioned in prompts.

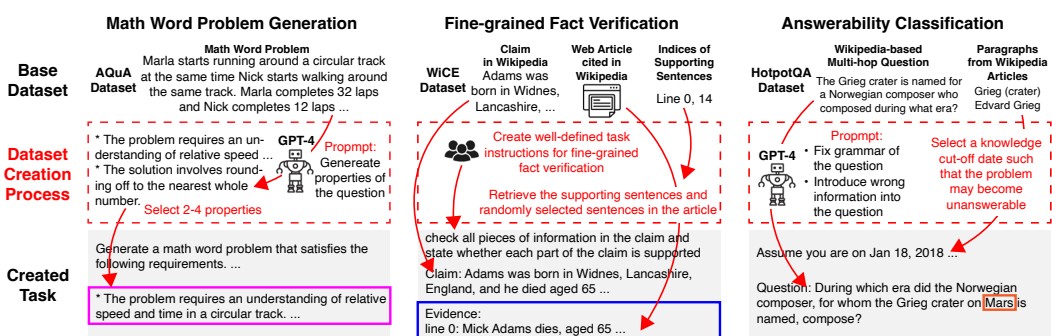

Figure 3: Creation processes of three tasks in *ReaLMistake*. Details are in Appendix B.

- **Context-Faithfulness**: Responses should be faithful to the context provided in inputs and should not ignore any part of the context. If the input instructs not to use any resources other than the provided context, the response should not be affected by parameterized knowledge (Zhou et al., 2023b). This criterion is often used as an objective "factuality" criterion for text summarization (Fabbri et al., 2021; Cao & Wang, 2021; Laban et al., 2023).

- **Parameterized Knowledge**: Responses should be factually correct. In this work, we focus on parameterized knowledge without allowing LLMs to access external resources. Prior studies also annotate mistakes in parameterized knowledge in a binary manner (Manakul et al., 2023; Min et al., 2023), supporting the objectiveness of this criterion.

**(2) Create Tasks Challenging for LLMs but Feasible for Humans.** We also highlight the difficulty in creating challenging tasks for recent strong LLMs. Classic objective tasks, such as mathematical reasoning at the grade school level, are too easy for recent LLMs (OpenAI, 2023; Anthropic, 2024) and we cannot collect mistakes. However, tasks that involve expert knowledge (Malaviya et al., 2023; Rein et al., 2023) or very long context (Wu et al., 2021; Pang et al., 2022) are not suitable for our purpose because it is too difficult for human annotators and users of this benchmark to conduct an analysis of error detection methods. Since it is challenging to create difficult tasks, some datasets include artificially created negative cases created by prompting LLMs to intentionally make mistakes (Zeng et al., 2024; Li et al., 2023a; Laban et al., 2023), but artificial mistakes cannot evaluate error detection in realistic settings.

To address these problems, we design tasks involving properties that are observed to be challenging for LLMs: (a) following requirements (Liu et al., 2024; Zhou et al., 2023a), (b) comparison of multiple texts (Laban et al., 2023; Chen et al., 2023b), and (c) detection of small mistakes in text (Cohen et al., 2023). As shown in Table 2, **tasks in our benchmark make GPT-4 introduce errors in more than 50% of cases, while their inputs consist of a few hundred words that only involve high-school-level mathematics and knowledge in Wikipedia**. These tasks allow us to efficiently collect errors made by LLMs.

### 3.2 Tasks

Our benchmark includes three challenging, objective, and realistic tasks designed for collecting diverse categories of errors in LLM responses. As discussed in the previous section, these tasks are designed so that they can be evaluated by the four objective and diverse criteria, eliminating the subjectivity in the error annotation process while ensuring the diversity. In addition, the three tasks are selected to involve skills required in real-world applications of LLMs (e.g., arithmetic reasoning, fact verification, and factual knowledge), in contrast to the objective tasks in prior work that have properties different from those of real-world uses of LLMs, such as chess (Wu et al., 2024) and word sorting (Tyen et al., 2024). Figure 2 includes an example from each task. Figure 3 visualizes the creation processes.

MathGen - **Math Word Problem Generation** is a task of generating math word problems that follow multiple requirements. This task is designed to be evaluated on *Reasoning Correctness* and *Instruction-Following*. This task instructs models to generate a question that

| Category of Requirements in *MathGen* Inputs | % |
|---|---|
| The problem requires an understanding of the concept of ... | 54.2 |
| The problem should include [price of ..., number of ...] | 45.8 |
| The problem should include [specific number] | 30.0 |
| The problem should include the phrase "...". | 23.7 |
| The problem should include [integers, fractions, etc.] and should not include any other type of number | 25.3 |
| The solution should involve [addition, linear equation, etc.] | 60.0 |
| The solution should include the phrase "...". | 18.9 |
| The solution should include the question "...". | 15.8 |
| The solution should include [integers, fractions, etc.] and should not include any other type of number | 63.2 |

Table 3: *Math Word Problem Generation* task consists of diverse requirements in 9 categories.

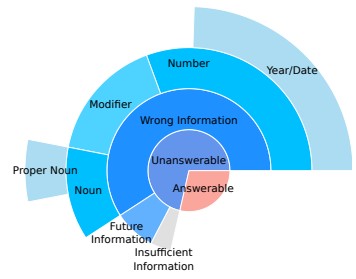

Figure 4: Distribution of questions in *Answerability Classification* task. This task includes diverse types of unanswerable questions.

follows all requirements in input (*Instruction-Following*) and also instructs to generate a solution for the generated question (*Reasoning Correctness*). Creating valid mathematical questions is challenging for LLMs and allows us to collect mistakes by LLMs without using advanced mathematical concepts, which are too difficult for annotators and users. **Creation Process:** We generate the requirements in inputs by using questions in AQuA (Ling et al., 2017). Specifically, we use GPT-4 to generate nine properties (e.g., topics, mathematical concepts, types of numbers; listed in Table 3) of each pair of a question and solution in AQuA and select two to four properties as requirements for the math word problem generation task. To make the task objective, we only include objective requirements.

FgFactV - **Fine-grained Fact Verification** is a task of checking whether each piece of information in a claim sentence is supported by the provided evidence. Fine-grained fact verification that instructs to provide detailed explanations has not been explored and similar tasks have been reported to be challenging for LLMs (Fabbri et al., 2021). This task is designed to be evaluated on *Reasoning Correctness* and *Context-Faithfulness*. This task instructs the models to provide reasoning (*Reasoning Correctness*) on whether each piece of information in the claim is supported by the evidence or not (*Context-Faithfulness*). To make sure that annotators can objectively evaluate LLM responses in a binary manner, we provide detailed instructions such as "*check all pieces of information in the claim and state reasoning on whether each part of the claim is supported*". **Creation Process:** We use WiCE (Kamoi et al., 2023) as a base dataset, which includes a factuality classification task. We reconstruct this task into fine-grained verification, which instructs models to explicitly check all pieces of information in the claim. Since WiCE includes very long web articles as evidence, we retrieve part of them to make the model inputs in our task shorter.

AnsCls - **Answerability Classification** is a task of classifying the answerability of factual questions. Answerability classification is a popular task (Rajpurkar et al., 2018; Kwiatkowski et al., 2019), but the performance of LLMs on challenging cases is unexplored. Our task instructs the models to answer the provided question, but they should classify the questions as unanswerable if the questions include factual errors. Unlike classic datasets that include entirely unanswerable questions, our task requires detecting small factual mistakes. This task is designed to be evaluated on *Reasoning Correctness* and *Parameterized Knowledge*. This task instructs models to reason whether the questions are answerable (*Reasoning Correctness*) by checking the factual correctness of questions (*Parameterized Knowledge*). **Creation Process:** We use HotpotQA (Yang et al., 2018) as a base dataset. To create unanswerable questions, we (1) add a requirement to use knowledge before a specific date to make the questions time-sensitive (Chen et al., 2021) and (2) use GPT-4 to introduce factual mistakes in the numbers, nouns, and modifiers in the questions. By using multiple prompts to introduce mistakes, we create unanswerable questions in diverse categories, as shown in Figure 4.

### 3.3 Annotation

14 faculty and graduate students participate in the annotation process. We train the annotators on multiple practice annotations. Each instance is annotated by one annotator and verified by another annotator. The tasks in our benchmark require annotators to carefully read the whole text in inputs and responses, and the annotations take about 90 hours in total (6 minutes for each case). Details are in Appendix C. **Filtering:** We remove pairs of input and output that involve errors detected by other criteria (e.g., grammar mistakes) or subjectivity in determining whether the response is erroneous or not. However, this process removes only 3.8% of cases, showing that our tasks are properly designed to be evaluated objectively by the four criteria. **Human Agreement:** To assess the human agreement and annotation quality, we evaluate the expert performance on 35 instances from each dataset ($35 \times 6$ instances in total). The average performance of three annotators on the three tasks was 95.7 in F1, verifying that humans can annotate binary error labels with high agreement.

## 4 Error Detection Performance on *ReaLMistake*

We evaluate 12 LLMs on the error detection task in *ReaLMistake*. The questions addressed in this section include: (1) How well can LLMs detect errors made by LLMs? (2) Can LLM-based error detectors provide reliable explanations for mistakes in LLM responses? (3) How can we improve LLM-based error detectors? and (4) Does *ReaLMistake* include challenging and diverse error detection tasks?

The LLM-based error detectors evaluated in our experiments use the same prompts of "classify whether the response includes errors" for all tasks, differing from criterion-specific detectors in prior work, such as factuality evaluators. As in Figure 1, evaluation of the criteria-independent detectors is enabled by the tasks in *ReaLMistake*, which do not involve subjectivity in the evaluation. In contrast, for example, the text summarization task cannot objectively evaluate criteria-independent detectors because they involve the subjective "relevance" criterion, although it can objectively evaluate factuality evaluators.

**LLMs for Detectors.** **Open:** Gemma 7B (Google, 2024), Llama 2 (13B and 70B) (Touvron et al., 2023), Mistral 7B (Jiang et al., 2023), Mixtral 8x7B (Jiang et al., 2024), Qwen 1.5 (14B and 72B) (Bai et al., 2023; Qwen Team, 2024), **Closed:** Gemini 1.0 Pro (Gemini Team Google, 2023), Claude 3 Opus (Anthropic, 2024), GPT-3.5 Turbo (Brown et al., 2020; Ouyang et al., 2022), and GPT-4 (2023-0613 and 2024-0125) (OpenAI, 2023). Details are in Appendix H.

**Prompts.** We use four zero-shot prompts (Kojima et al., 2022) for error detection that have variations in two properties: different **wordings** and flipped **order** of the binary error label options (Section 4.3, Figure 7). In Table 4, we use the average performance of the four prompts to alleviate the influence of biases caused by the prompt design.

### 4.1 Strong LLMs Detect Errors in LLM Responses at Low Recall

Table 4 shows the error detection performance of 12 LLMs with zero-shot prompts on *ReaLMistake*. It shows that LLM-based error detectors often perform worse than random baselines in F1, showing that **the error detection task in *ReaLMistake* is difficult even for strong LLMs such as Claude 3 and GPT-4**. The results also show that the three tasks in *ReaLMistake* exhibit different difficulties. GPT-4 achieves reasonable F1 scores on *MathGen* but performs poorly on *FgFactV* and *AnsCls* with very low recall. This result suggests that error detection on tasks with different properties has varying difficulties, and indicates that *ReaLMistake* **provides error detection tasks on diverse tasks with different properties**.

To analyze relationships between the performance of LLMs on error detection and other popular tasks, we compare the performance of 12 LLMs on *ReaLMistake* to LMSYS Elo Rating (Zheng et al., 2023).[3] Figure 5 shows that **LLMs with higher LMSYS Elo Rating (stronger models) detect errors in LLM responses with higher precision but with lower recall**, revealing the challenges in LLM-based error detection: low recall of strong LLMs and

---

[3]`https://huggingface.co/spaces/lmsys/chatbot-arena-leaderboard` (March 16, 2024)

| Error Detector | | Gemma 7B | Llama 2 13B | Llama 2 70B | Mistral 7B | Mistral 8x7B | Qwen 1.5 14B | Qwen 1.5 72B | GPT3.5 0125 | Gemini 1.0 Pro | Claude3 Opus | GPT-4 0613 | GPT-4 0125 | Random | Expert Human |
|---|---|---|---|---|---|---|---|---|---|---|---|---|---|---|---|
| **F1** | | | | | | | | | | | | | | | |
| GPT-4 0613 | MathGen | 46.5 | 54.2 | 59.5 | 6.9 | 45.5 | 52.3 | 32.8 | 65.3 | 42.5 | 50.1 | 63.1 | **70.9** | 62.1 | 90.0 |
| | FgFactV | 60.3 | 65.4 | **69.9** | 50.9 | 46.8 | 57.7 | 24.9 | 41.4 | 45.8 | 48.9 | 12.7 | 20.8 | 62.9 | 95.5 |
| | AnsCls | 59.2 | 69.8 | **69.8** | 48.1 | 38.3 | 53.8 | 15.1 | 28.8 | 40.7 | 38.5 | 20.0 | 22.1 | 62.1 | 90.5 |
| Llama 2 70B | MathGen | 54.3 | 56.6 | 69.2 | 9.0 | 56.0 | 54.9 | 50.3 | 72.3 | 52.9 | 81.8 | 88.7 | **90.8** | 80.0 | 98.3 |
| | FgFactV | 68.9 | 78.7 | **81.8** | 68.2 | 35.1 | 64.6 | 18.3 | 34.2 | 42.0 | 45.2 | 38.8 | 68.5 | 80.6 | 100.0 |
| | AnsCls | 34.8 | **77.4** | 51.6 | 61.9 | 29.8 | 44.9 | 5.1 | 3.7 | 16.4 | 23.2 | 61.6 | 75.9 | 81.2 | 100.0 |
| **Precision** | | | | | | | | | | | | | | | |
| GPT-4 0613 | MathGen | 61.6 | 62.6 | 73.0 | 22.8 | 75.5 | 77.4 | 82.9 | 77.3 | 78.1 | **94.9** | 94.4 | 88.9 | 62.1 | 100.0 |
| | FgFactV | 62.3 | 62.0 | 62.4 | 58.4 | 61.3 | 59.8 | 67.1 | 49.9 | 67.2 | 78.2 | **100.0** | 95.0 | 62.9 | 95.5 |
| | AnsCls | 64.0 | 62.2 | 65.2 | 59.8 | 60.9 | 68.6 | 55.4 | 72.8 | 78.4 | 74.9 | 79.9 | **88.2** | 62.1 | 95.0 |
| Llama 2 70B | MathGen | 82.6 | 79.5 | 88.6 | 41.8 | 89.0 | 96.2 | 94.5 | 86.4 | 90.0 | 95.0 | **97.7** | 95.2 | 80.0 | 100.0 |
| | FgFactV | 83.5 | 81.9 | 82.4 | 80.0 | 96.3 | 83.2 | 73.7 | 98.7 | 85.7 | **99.3** | 85.4 | 92.6 | 80.6 | 100.0 |
| | AnsCls | 80.5 | 82.5 | 77.3 | 83.8 | 86.3 | 74.8 | 70.5 | 69.4 | 78.3 | **100.0** | 97.1 | 98.4 | 81.2 | 100.0 |
| **Recall** | | | | | | | | | | | | | | | |
| GPT-4 0613 | MathGen | 50.0 | 52.3 | **75.3** | 4.3 | 35.1 | 49.7 | 23.3 | 64.1 | 41.7 | 35.9 | 48.0 | 59.5 | 62.1 | 81.8 |
| | FgFactV | 60.5 | 73.0 | **83.2** | 45.2 | 44.3 | 60.8 | 17.0 | 36.9 | 39.2 | 38.6 | 6.8 | 11.9 | 62.9 | 95.5 |
| | AnsCls | 57.2 | **81.3** | 79.3 | 45.4 | 29.6 | 54.0 | 8.9 | 19.3 | 31.6 | 26.4 | 11.5 | 12.6 | 62.1 | 86.4 |
| Llama 2 70B | MathGen | 51.2 | 50.2 | 72.9 | 5.7 | 44.3 | 47.3 | 37.5 | 65.8 | 46.9 | 72.7 | 81.2 | **86.9** | 80.0 | 96.7 |
| | FgFactV | 61.8 | 77.5 | **82.9** | 60.7 | 24.4 | 61.2 | 11.0 | 24.2 | 32.2 | 32.6 | 25.8 | 54.8 | 80.6 | 100.0 |
| | AnsCls | 23.3 | **77.5** | 46.7 | 52.3 | 19.4 | 45.2 | 2.7 | 1.9 | 9.8 | 13.3 | 45.2 | 62.1 | 81.2 | 100.0 |

Table 4: Error detection performance of 12 LLMs with zero-shot prompts on *ReaLMistake*. This table shows the average performance on four prompts in Section 4.3. "Random" baseline predicts each instance as an error in the same probability as the frequency of the error labels for each dataset. Human performance is evaluated on 35 cases in each setting. Gray color represents the values worse than the random baseline.

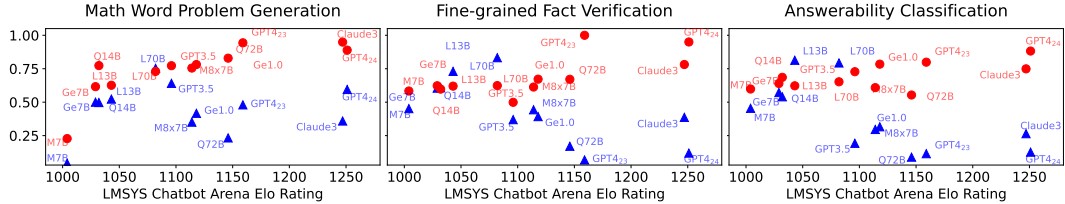

Figure 5: Relationship between LMSYS Elo Rating (Zheng et al., 2023) and error detection performance on GPT-4 responses in *ReaLMistake* of 12 LLMs. Stronger LLMs detect errors made by GPT-4 with higher precision ● but with lower recall ▲. Comparison to the performance on MMLU (Hendrycks et al., 2021) in Appendix G.1 shows the same trends.

low precision of weak LLMs. Comparison to the performance on MMLU (Hendrycks et al., 2021), which shows consistent results, and correlation coefficients are in Appendix G.1.

## 4.2 Explanations Generated by LLM-based Error Detectors are Not Reliable

Our prompts instruct LLM-based error detectors to provide an explanation in addition to predicting the binary error label. We manually analyze the explanations provided by error detectors based on GPT-4-0125, Claude 3 Opus, and three open-source models that show relatively good binary error detection performance in Table 4

Figure 6 shows the distribution of mistakes in the explanations by the error detectors on responses from GPT-4-0613. We observe that GPT-4-0125 and Claude 3 Opus generate explanations of relatively good quality, but they still make mistakes, especially on *MathGen*. **Explanations by open-source models are more often wrong even when the binary predictions are correct**. They often make mistakes even for correct responses from GPT-4-0613, not just missing errors, resulting in low precision. To improve the error detection performance,

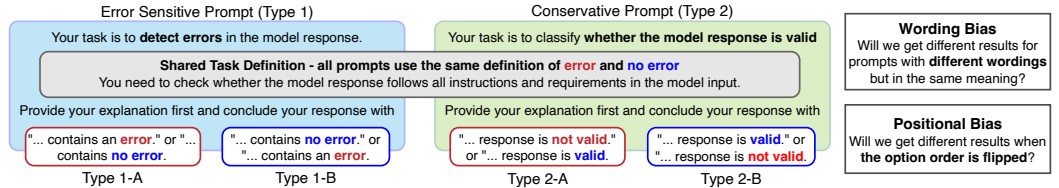

Figure 6: Distribution (%) of mistakes in explanations by LLM-based error detectors (Type 1-A) on responses from GPT-4-0613 in *ReaLMistake*. Open-source models introduce more mistakes compared to GPT-4 and Claude 3. We manually classify outputs from error detectors for 20 cases in each dataset: correct binary label prediction with ● correct, ◔ insufficient (some requirements not evaluated), ◑ wrong reasoning, or ○ no reasoning; wrong binary label prediction with ⊘ no reasoning, ◔ insufficient (missing errors), or ● wrong reasoning. Appendix I provides example outputs from LLM-based error detectors.

Figure 7: Prompts for LLM-based error detectors for LLM responses. We evaluate four zero-shot prompts: two different wordings (Type 1 and 2) and the order of the error label options (Type A and B). We use a shared task definition in all prompts to make sure that they use the same definition of errors. The full prompts are provided in Appendix E.

this analysis suggests that we need to make strong LLMs more careful about the mistakes in the provided LLM responses and improve the reasoning capability of open-source models.

### 4.3 Error Detection is Sensitive to Prompt Design, but Not Easy to Improve Performance

We analyze the influence of two small changes in prompts on error detection by comparing the performances of LLM-based error detectors with four prompts (Figure 7). **Positional Bias** (Wang et al., 2023a; Zheng et al., 2023; Pezeshkpour & Hruschka, 2023)**:** We evaluate the influence of the order of the error label options in the prompts. In an average of 12 LLMs and 3 tasks, putting the "error" option first (Type B in Figure 7) has $16.0 \pm 21.7\%$ (Type 1) and $27.2 \pm 23.9\%$ (Type 2) higher recall, suggesting that LLM-based error detectors prefer the first option. **Wording Bias:** We compare two wordings in error detection that ask to "detect errors in the response" (Type 1) and "evaluate whether the response is valid" (Type 2) while providing the same definition to make sure that they are semantically identical. In an average of 12 LLMs and 3 tasks, Type 1 has $16.9 \pm 20.3\%$ higher recall.[4] These results show that **recall of error detection is sensitive to small changes in prompts**.[5]

However, we also observe that these biases cannot be easily used to improve error detectors. First, although low recall is a problem in error detection by strong LLMs, we observe that small changes in prompts have a very small influence on the recall of strong models. In addition, we do not observe consistent trends in precision, although low precision is a problem in error detection by weaker models. These results suggest that non-trivial improvements to prompts or frameworks are needed to improve the error detection performance.

### 4.4 Popular Techniques to Improve LLMs Do Not Improve LLM-based Error Detection

A possible approach to improve LLM-based error detectors is to apply techniques proposed for improving LLM-based evaluators for LLM responses (e.g., pairwise evaluation). We

---

[4]We compare the average performance of Type 1-A and 1-B with that of Type 2-A and 2-B.

[5]The numbers after "±" represent standard deviation. Detailed results are in Appendix G.2.

apply three popular approaches to LLM-based error detection: self-consistency (Wang et al., 2023b), integrating responses by multiple LLMs (Zhang et al., 2023; Li et al., 2023b; Cohen et al., 2023; Chan et al., 2024), and providing evaluation steps in prompts (Liu et al., 2023a).

Specifically, we evaluate three simple methods: self-consistency with five samples, the majority vote of binary error predictions by multiple models, and providing human-written instruction of evaluation steps in prompts. **However, none of these methods in popular approaches improves error detection performance** (Appendix A). This observation indicates that applying existing techniques for LLM-based evaluators is not sufficient to improve LLM-based error detectors, and *ReaLMistake* provides challenging error detection tasks.

## 5 Future Directions - Evaluation on Ambiguous Errors

In this work, we propose to evaluate error detectors on objective errors because this approach has multiple advantages, including smaller noises, the absence of subjectivity of annotators, binary annotations, and smaller annotation costs. However, since many NLP tasks are subjective, we expect future research to evaluate whether error detectors have a good correlation with human evaluators on ambiguous errors. A possible approach to collecting benchmarks for evaluating error detectors on ambiguous errors is to collect annotations by multiple annotators for each instance to get average error scores. However, difficulties in creating such benchmarks include (1) hiring annotators who represent the distribution of (possibly subjective) error annotations by general people, (2) evaluating binary predictions from LLM-based error detectors using labels by multiple annotators, and (3) high annotation cost. We expect future work to address these challenges.

## 6 Conclusion

We introduce *ReaLMistake*, the first benchmark for evaluating error detection methods for LLM responses, consisting of objective, realistic, and diverse errors made by LLMs. Our experiments on this benchmark with error detectors based on 12 LLMs show that detecting mistakes in LLMs (GPT-4 and Llama 2 70B) is challenging even for recent LLMs.

## Reproducibility Statement

**Experiments.** We provide a GitHub repository[6] that includes the *ReaLMistake* benchmark (also hosted at Hugging Face Datasets[7]), all outputs from LLM-based error detectors used in Section 4, and evaluation code.

**Annotation.** Appendix B provides annotation instructions we used for our benchmark.

## Acknowledgment

This work was supported by a Cisco Research Grant. We thank Greg Durrett for the valuable discussions and comments. We appreciate valuable suggestions from anonymous reviewers.

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

# Table of Contents of Appendix

# A   Performance of Advanced Error Detection Methods on *ReaLMistake*

This section includes the results of the advanced error detection methods in Section 4.4. In our experiments, the three methods do not improve the error detection performance.

## A.1   Self Consistency

Table 5 shows the error detection performance of three LLM-based error detectors (prompt type 1-A in Figure 7) with self-consistency (Wang et al., 2023b) on *ReaLMistake*. We use $k = 5$ and temperature=0.5 for all models. We do not observe improvement compared to the performance of a single output with temperature=0.0.

## A.2   Majority Vote

Table 6 shows the performance of the majority vote on binary predictions from LLM-based detectors based on three open-source LLMs. We use four prompts in Figure 7 and take the majority label of 12 outputs (3 models × 4 prompts). We do not observe improvement in the approach of taking the majority vote on the binary predictions.

## A.3   Human-written Instruction of Evaluation Steps

The idea of providing instructions on evaluation steps is motivated by G-Eval (Liu et al., 2023b), which provides LLM-generated evaluation steps to LLM-based evaluators. For simplicity, we manually create human-written evaluation steps and provide the following instructions in the error detection prompt to help error detectors provide better reasoning. We added the following instructions at the end of the Prompt 1-A (Appendix E).

---

**Human-written Instruction of Evaluation Steps**

Follow the instructions below to check whether the model response contains an error:
1. Read the model input carefully.
2. Read the model response carefully.
3. Check whether the model response follows all instructions and requirements in the model input. Provide your explanation on each requirement.
4. Select your answer from "Therefore, the model response contains an error." or "Therefore, the model response contains no error.

---

Table 7 includes the improvements made by providing human-written instruction of evaluation steps in prompts. We do not observe any consistent improvement by this method.

| Detector | Mixtral 8x7B | | GPT-3-Turbo | | GPT-4-0125 | |
|---|---|---|---|---|---|---|
| Temperature | T=0.0 | T=0.5 | T=0.0 | T=0.5 | T=0.0 | T=0.5 |
| # sample | k=1 | k=5 | k=1 | k=5 | k=1 | k=5 |
| F1 | | | | | | |
| MathGen | 63.0 | 64.2 | 81.2 | 82.4 | 74.5 | **83.0**[*] |
| FgFactV | 82.4 | 75.4 | 67.9 | 61.8 | 22.9 | 27.8 |
| AnsCls | 41.7 | 44.0 | 52.2 | 51.2 | 23.5 | 18.2 |
| Precision | | | | | | |
| MathGen | 68.0 | 70.8 | 70.0 | 71.8 | 86.4 | 91.7 |
| FgFactV | 75.7 | 68.4 | 76.0 | 70.8 | 100.0 | 100.0 |
| AnsCls | 50.0 | 50.0 | 66.7 | 73.3 | 66.7 | 60.0 |
| Recall | | | | | | |
| MathGen | 58.6 | 58.6 | 96.6 | 96.6 | 65.5 | **75.9**[*] |
| FgFactV | 90.3 | 83.9 | 61.3 | 54.8 | 12.9 | 16.1 |
| AnsCls | 35.7 | 39.3 | 42.9 | 39.3 | 14.3 | 10.7 |

Table 5: Performance of LLM-based error detectors (prompt type 1-A) with **self-consistency** (k=5, temperature=0.5) on 50 cases from three datasets with GPT-4-0613 responses in *ReaLMistake*. Self-consistency does not provide consistent improvement in the error detection task in *ReaLMistake*. [*]: $p < 0.1$ on the pairwise bootstrap test with 1,000 samples.

| | | Llama 2 70B | Mixtral 8x7B | Qwen 1.5 72B | Majority Vote of Three Models |
|---|---|---|---|---|---|
| | | F1 | | | |
| GPT-4 0613 | MathGen | 59.5 | 45.5 | 32.8 | 37.9 |
| | FgFactV | 69.9 | 46.8 | 24.9 | 39.1 |
| | AnsCls | 69.8 | 38.3 | 15.1 | 26.8 |
| Llama 2 70B | MathGen | 69.2 | 56.0 | 50.3 | 58.1 |
| | FgFactV | 81.8 | 35.1 | 18.3 | 22.1 |
| | AnsCls | 51.6 | 29.8 | 5.1 | 1.5 |
| | | Precision | | | |
| GPT-4 0613 | MathGen | 73.0 | 75.5 | 82.9 | 75.9 |
| | FgFactV | 62.4 | 61.3 | 67.1 | 62.5 |
| | AnsCls | 65.2 | 60.9 | 55.4 | 60.0 |
| Llama 2 70B | MathGen | 88.6 | 89.0 | 94.5 | 93.1 |
| | FgFactV | 82.4 | 96.3 | 73.7 | 100.0 |
| | AnsCls | 77.3 | 86.3 | 70.5 | 33.3 |
| | | Recall | | | |
| GPT-4 0613 | MathGen | 75.3 | 35.1 | 23.3 | 25.3 |
| | FgFactV | 83.2 | 44.3 | 17.0 | 28.4 |
| | AnsCls | 79.3 | 29.6 | 8.9 | 17.2 |
| Llama 2 70B | MathGen | 72.9 | 44.3 | 37.5 | 42.2 |
| | FgFactV | 82.9 | 24.4 | 11.0 | 12.4 |
| | AnsCls | 46.7 | 19.4 | 2.7 | 0.8 |

Table 6: Performance of the majority vote of binary predictions from 3 LLM-based error detectors (rightmost column). The performance of each model (left columns) is the average performance of the four prompts (Table 4), and the majority vote is taken over 12 outputs (3 models × four prompts). Yello color represents values better than the performances of majority vote predictions. We do not observe improvement by taking the majority vote of predictions from multiple models.

| Error Detector | | Gemma 7B | Llama 2 13B | Llama 2 70B | Mistral 7B | Mistral 8x7B | Qwen 1.5 14B | Qwen 1.5 72B | GPT3.5 0125 | Gemini 1.0 Pro | Claude3 Opus | GPT-4 0613 | GPT-4 0125 |
|---|---|---|---|---|---|---|---|---|---|---|---|---|---|
| | | | | | | | **F1** | | | | | | |
| GPT-4 0613 | MathGen | 17.0 | -28.4 | -0.2 | 11.2 | 4.3 | -4.2 | 15.4 | -2.0 | -29.4 | 14.3 | 2.3 | -1.0 |
| | FgFactV | 4.6 | -1.9 | -0.2 | -8.7 | 2.8 | -6.0 | 17.1 | -10.4 | -15.0 | -29.5 | -3.9 | 3.3 |
| | AnsCls | 2.4 | -2.0 | 1.4 | -5.9 | 1.1 | -6.8 | 0.0 | -14.5 | -29.7 | -8.5 | -5.3 | 6.1 |
| Llama 2 70B | MathGen | 18.4 | -30.9 | 0.7 | -8.9 | 3.5 | -4.1 | 3.6 | -4.5 | -24.7 | 4.2 | -0.5 | -0.1 |
| | FgFactV | 1.9 | -2.9 | 0.4 | -23.3 | 13.4 | -53.9 | 9.7 | -39.3 | -20.2 | 5.4 | -5.8 | -7.0 |
| | AnsCls | 3.2 | -11.0 | 4.7 | -17.0 | -19.0 | -44.8 | 10.2 | -5.6 | -20.1 | -5.9 | -2.7 | -0.8 |
| | | | | | | | **Precision** | | | | | | |
| GPT-4 0613 | MathGen | 17.1 | -13.7 | 0.2 | 17.2 | 5.6 | 5.6 | -6.0 | 2.5 | 6.6 | -6.9 | 1.9 | 6.1 |
| | FgFactV | 0.2 | 3.6 | 0.2 | -1.8 | -1.4 | 2.2 | 9.5 | 3.0 | 4.3 | -6.7 | 0.0 | 0.8 |
| | AnsCls | -4.3 | 1.8 | -0.5 | -3.0 | 13.8 | 2.1 | 0.0 | -3.4 | -10.5 | -1.8 | -5.0 | -5.7 |
| Llama 2 70B | MathGen | 12.9 | -0.5 | 0.7 | 25.0 | 8.1 | 1.0 | -1.1 | 3.3 | 12.0 | 1.3 | 0.8 | 1.4 |
| | FgFactV | -3.2 | 5.7 | 0.1 | 11.8 | 2.7 | -13.2 | 1.2 | 1.3 | 8.7 | 1.7 | 4.8 | -3.5 |
| | AnsCls | -4.1 | -0.8 | -3.6 | -1.6 | -3.0 | -1.4 | 12.6 | -2.8 | 1.0 | 0.0 | 1.5 | 1.1 |
| | | | | | | | **Recall** | | | | | | |
| GPT-4 0613 | MathGen | 16.1 | -36.8 | -1.1 | 8.0 | 3.4 | -19.5 | 31.0 | -9.2 | -47.1 | 19.5 | 2.3 | -4.6 |
| | FgFactV | 12.5 | -12.5 | -1.1 | -12.5 | 10.2 | -17.0 | 19.3 | -20.5 | -25.0 | -31.8 | -2.3 | 2.3 |
| | AnsCls | 11.5 | -10.3 | 5.7 | -10.3 | -3.4 | -20.7 | 0.0 | -12.6 | -32.2 | -8.0 | -3.4 | 4.6 |
| Llama 2 70B | MathGen | 18.8 | -43.0 | 0.8 | -7.8 | 0.8 | -9.4 | 6.2 | -11.7 | -39.8 | 6.2 | -1.6 | -1.6 |
| | FgFactV | 7.0 | -13.2 | 0.8 | -36.4 | 15.5 | -70.5 | 8.5 | -38.8 | -26.4 | 5.4 | -5.4 | -7.8 |
| | AnsCls | 3.8 | -22.3 | 14.6 | -23.8 | -13.8 | -65.4 | 6.2 | -3.1 | -13.8 | -3.8 | -3.1 | -1.5 |

Table 7: Improvement of LLM-based error detectors by using human-written instructions of evaluation steps. This table compares the performance of the original prompt type 1-A in Figure 7 and the updated prompt type 1-A with human-written evaluation steps. The gray color represents that human-written instructions decrease performance. This approach does not provide improvement from the original simple prompt.

# B   Dataset Creation Process

We provide details of the dataset creation process (Figure 3) for tasks in *ReaLMistake* (Section 3.2). Please refer to Appendix I for full inputs of these tasks.

### B.1   Math Word Problem Generation

Math Word Problem Generation is a task to create math word problems that follow the provided requirements. We use math word problems and solutions in the AQuA dataset (Ling et al., 2017) to generate requirements. We generate 9 types of requirements to introduce diversity to this task (Table 3). First, to create requirements about the types of numbers (e.g., integers, fractions) in questions and solutions, we create a data type detection script using regular expression operations. Second, We use the following prompt to generate 7 properties of the question in AQuA by using GPT-4-0613. We note that this prompt only asks GPT-4 to retrieve specified information from the questions, so we believe that there is no strong influence caused by the choice of LLMs in this process.

---

Your task is to generate properties of the provided math word question.
* Each generated sentence should be one sentence long.
* You do not need to generate formatting information such as "multiple options are provided as potential answers".
* The generated properties should not include "like" or "such as".
* You should only generate a list of properties. Do not generate anything else.

The generated properties should be in the following format:
* The first one should explain the overview of the question.
* The second one should explain the instances included in the question.
* The third one should include a short key phrase extracted from the question. The key phrase should not include more than one number.
* The fourth one should include only one randomly selected number in the question. This number should not be included in the key phrase.
* The fifth one should explain mathematical operations (e.g., linear equation) included in the solution. When there are unknown variables in equations, specify what they represent.
* The sixth one should include a short key phrase extracted from the solution. The key phrase should not include any number. This should not include the answer to the question.
* The seventh one should include a randomly selected equation or calculation in the solution. You need to insert spaces to improve readability if necessary. This should not include the answer to the question.

[few-shot examples]

---

### B.2   Fine-grained Fact Verification

We use the WiCE dataset (Kamoi et al., 2023) as a base dataset. WiCE includes a fact-verification classification task and each instance of the dataset includes a claim (sentence) in a Wikipedia article, web articles cited for the claim in Wikipedia, an entailment label (supported, partially supported, or not supported), and indices of supporting sentences (a part of the web articles supporting the claim).

Web articles in WiCE are too long for our benchmark, so we only use a part of them in our tasks. Specifically, we use the oracle chunks provided in WiCE, which include the supporting sentences in the web articles and randomly selected sentences from web articles.[8]

We note that we changed the entailment classification task to a binary classification of supported or not supported. In our task, the original partially supported cases are regarded

---

[8]https://github.com/ryokamoi/wice/tree/main/code_and_resources/entailment_inputs/oracle_chunks

as not supported. We added instructions to perform fact verification for all pieces of information in the claim to reconstruct the task to fine-grained fact verification.

### B.3 Answerability Classification

Answerability Classification is a task to classify questions as unanswerable if they include factual errors or any other problems. We use the HotpotQA dataset (Yang et al., 2018) as a base dataset. Each instance in HotpotQA includes a multi-hop question about knowledge in Wikipedia articles and paragraphs from Wikipedia articles that include information required to answer the question. The original task in HotpotQA is a mult-hop QA using the provided evidence, but we do not provide the paragraphs and make it a closed-book QA task.

First, since questions in HotpotQa are often grammatically unnatural, we use GPT-4-0613 to improve the questions without changing any properties.

---

Your task is to improve the following question while keeping the original meaning of the question. You should only generate the rewritten question. Do not include anything else in your response.

[few-shot examples]

---

Second, we use three types of automatic processes to create diverse unanswerable questions (Figure 4). We use each of the following processes for 1/3 of the cases in our dataset.

**Factual Mistakes in Numbers.** The first type is mistakes in numbers. We provide the following prompt to GPT-4-0613 to generate (possibly) unanswerable questions that include factual mistakes in numbers.

---

We provide a question that can be answered by using Wikipedia articles.
First, add correct additional information from related Wikipedia articles to the question that includes at least one number (e.g., year, age, quantity). The added information should not change the answer to the question.
Second, introduce a minor mistake in the number in the added information.
Do not include anything else in your response.

[few-shot examples]

---

**Other Factual Mistakes.** The second type is mistakes other than numbers. We provide the following prompt to GPT-4-0613. We instruct not to make mistakes in proper nouns because they make the answerability detection task easier. However, this requirement is not always satisfied as shown in Figure 4.

---

We provide a question that can be answered by using Wikipedia articles.
First, add correct additional information from related Wikipedia articles to the question. The added information should not change the answer to the question.
Second, introduce a factual mistake in the added information (nouns, adjectives, verbs, etc.). However, do not introduce mistakes in people's names, titles of books or movies, and locations (e.g., countries).
You should introduce factual errors, so the mistakes should not use synonyms or paraphrases that do not change the factual correctness of the information.
Do not include anything else in your response.

[few-shot examples]

---

**Time Sensitive Questions.** The third type is time sensitivity. All inputs in our Answerability Classification include time constraints in a format of:

> Assume you are on {date} and questions that require knowledge after this date should be classified as unanswerable.

For the questions involving the above two types of factual mistakes, we randomly select dates between 2022-09-01 and the date the first version of the Wikipedia article was created (we select the newest article from the ones provided for each question in HotpotQA). However, randomly selected dates often do not make the questions unanswerable.

To create more questions that are unanswerable due to the time constraint, we use years in the paragraphs provided in HotpotQA as a knowledge resource for each question. Specifically, we randomly select years in the paragraphs provided for each question in HotpotQA and use the time constraint of one year before the selected year. This strategy makes at least one event in the paragraphs unanswerable, although it is not guaranteed that this event is involved in the question-answering process.

## C Annotation Instructions

Tasks in *ReaLMistake* include detailed task definitions in inputs (prompts for LLMs), and binary error detection on these tasks is well-defined without any additional instructions (please refer to Appendix I for examples of full inputs of our tasks). However, we provide annotation instructions to annotators to further improve the annotation quality and provide instructions on error category annotations. We provide 9-page annotation instructions with multiple annotation examples.

To make sure that annotators understand these instructions, we provide five challenging cases for each dataset ($3 \times 5$ cases in total) as a practice annotation. We manually check their annotations and provide detailed feedback. If they make major mistakes, we provide another five cases. We continue this process until all annotators clearly understand the annotation instructions. This phase took about two weeks in total to train all annotators.

This section provides key parts of the annotation instructions.

### C.1 Definition of Error Labels

- error
    - Select this option when the response includes at least one error.
    - Provide a detailed explanation in the "feedback" column.
    - Responses with any kind of errors, not restricted to the provided error categories, should be categorized as errors (e.g., grammar mistakes). When you do not select any of the four categories, please explain in the "feedback" column.
- no_error
    - Select this option only when the response does not include any error
    - You do not need to provide feedback.
- not_sure
    - Select this option when you cannot determine the binary error label.
    - For example, you should select this option when mistakes in the output are too minor or the model input is invalid.
    - The cases with this label will be removed from the final dataset.

## C.2 Task-Specific Instructions

For analysis purposes, we categorize errors in model responses into four categories. Since model inputs do not provide any information about error categories, we provide detailed instructions on error category annotations.

**Math Word Problem Generation**

---

In this task, each model output includes a generated question and solution. We expect mistakes in this task to be mainly in Instruction-Following and Reasoning Correctness. In this task, an output from the models includes a generated question and solution.

- When the generated question or solution does not follow the constraints, select the Instruction-Following error category.
  - When the generated question is invalid (unanswerable), you also need to select the Instruction-Following error category because the instruction says "The generated question should be valid and answerable".
- When the solution includes wrong reasoning, select the Reasoning Correctness error category.
- Unnecessary information
  - When the question includes unnecessary information, it should be marked as the Instruction-Following error because the instruction says not to include unnecessary information in the generated questions.
  - In solution:
    * When the solution provides two possible ways to answer the question or verify the final answer, it is regarded as an Instruction-Following error because the instruction says not to include this.
    * When the solution includes unnatural and unnecessary steps other than the above cases, select the Reasoning Correctness error category.

[Multiple Annotation Examples]

---

**Fine-grained Fact-Checking**

- Check whether all pieces of information in the claim are verified in the response. If any part of the claim is not verified in the response, mark it as a Context-Faithfulness error even if the final answer is correct. Note that we regard the claim as context as well as the evidence in this task because this is a task to compare two texts (claim and evidence).
- Check whether each piece of information has been verified correctly.
  - If the response misses or misinterprets any part of the evidence, mark it as a Context-Faithfulness error.
  - If the response provides incorrect reasoning from correctly retrieved information, mark it as a Reasoning Correctness error.
- Check whether the final answer is faithful to the verification of each piece of information by the model. If not, mark it as an error of Reasoning Correctness.
- If the response does not follow the formatting requirements, mark it as an Instruction-Following error.

[Multiple Annotation Examples]

**Answerability Classification**

In this task, we provide factual questions with time constraints that may include incorrect information. LLMs need to classify the question as unanswerable if (1) it requires information after the specified date or (2) it includes incorrect information. Many questions in this task are unanswerable, but some questions may still be answerable. You need to check all the information in each question by using Wikipedia or any other reliable resources.

- If the question includes mistakes but the model response does not point out any factual mistakes, this is a Parameterized Knowledge error.
  - When there are multiple mistakes in a question, the response only needs to point out one mistake. The response does not need to point out all mistakes.
  - We introduce an intentional error in the questions. However, the original question can already include mistakes.
- If the response includes factual mistakes, they are also errors in Parameterized Knowledge.
  - The response can include factual mistakes even when the answerability classification is correct. This case is also a Parameterized Knowledge error.
  - For example, the response can detect one mistake in the question correctly but include factual mistakes caused by another mistake in the question. The response only needs to detect one mistake to clarify it as unanswerable, but should not include factual mistakes.
- Mistakes in reasoning that do not involve factual information should be marked as a Reasoning Correctness error.
- If the response correctly points out the mistakes but does not classify the question as unanswerable, classify it as a Reasoning Correctness error.
- The question should be classified as unanswerable if it requires any information after the timestamp in the instruction. If the response ignores this instruction, it should be marked as an Instruction-Following error.
- If the response does not include any factual mistakes or reasoning mistakes but ignores factual mistakes in the question, this is an error of Instruction-Following.

[Multiple Annotation Examples]

## D   Diversity of Tasks in *ReaLMistake*

**Math Word Problem Generation.**   Table 3 includes the distribution of 9 categories of requirements included in this task.

**Fine-grained Fact Verification.**   Figure 8 shows the distribution of topics of Wikipedia articles used in the Fine-grained Fact Verification task.

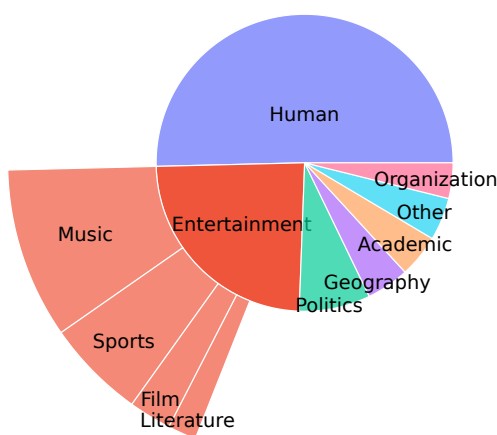

Figure 8: Topics of Wikipedia articles in the Fine-grained Fact Verification task.

**Answerability Classification Task.**   Figure 4 shows the distribution of the reasons for the unanswerability of the questions in this task.

# E  Prompts for LLM-based Error Detection

In the error detection experiments in Section 4, we use four simple zero-shot prompts described in Figure 7. The following is the full text of the prompts 1-A and 2-A. The green text represents different wordings (Type 1 and 2), and red and blue texts represent the options for the binary error label prediction (Type A and B). 1-B and 2-B flip the order of the options at the end of the prompts 1-A and 2-B. The sentences in black color in the first paragraph explain the shared definition of the binary error labels.

When we evaluate the binary classification of the LLM-based error detectors, we classify the responses as predicting "`error`" if they include a phrase "contains an error" or "response is not valid", and classify as predicting "`no_error`" if they include "contains no error" or "response is valid". Otherwise, the responses are classified as invalid. Therefore, we did not evaluate if the responses provide their answer at the end of their responses, which is often not satisfied by weaker models.

---

**Error Detection Prompt 1-A**

We provide an input to a language model (model input) and a response from the model (model response). Your task is to detect errors in the model response. You need to check whether the model response follows all instructions and requirements in the model input. You need to classify the model response as an error if the model response does not follow any of the instructions or requirements in the model input.

===== Model Input Begins =====

{input}

===== Model Input Ends =====

===== Model Response Begins =====

{response}

===== Model Response Ends =====

In your response, provide your explanation first and conclude your response with "Therefore, the model response contains an error." or "Therefore, the model response contains no error."

---

**Error Detection Prompt 2-A**

We provide an input to a language model (model input) and a response from the model (model response). Your task is to evaluate the quality of the model response and classify whether it is valid or not. You need to check whether the model response follows all instructions and requirements in the model input. You need to classify the model response as valid only if the model response follows all instructions and requirements in the model input.

===== Model Input Begins =====

[Omitted. Same as Prompt 1-A.]

===== Model Response Ends =====

In your response, provide your explanation first and conclude your response with "Therefore, the model response is not valid." or "Therefore, the model response is valid."

---

## F  Additional Results of Error Detection Performance

This section provides additional results for the experiments in Section 4. Table 8 shows the error detection performance in Table 4 in accuracy.

| Error Detector | | Gemma 7B | Llama 2 13B | Llama 2 70B | Mistral 7B | Mistral 8x7B | Qwen 1.5 14B | Qwen 1.5 72B | GPT3.5 0125 | Gemini 1.0 Pro | Claude3 Opus | GPT-4 0613 | GPT-4 0125 | Random | Expert Human |
|---|---|---|---|---|---|---|---|---|---|---|---|---|---|---|---|
| | | | | | | Accuracy | | | | | | | | | |
| GPT-4 0613 | MathGen | 43.4 | 47.3 | 58.2 | 36.1 | 50.7 | 56.1 | 48.2 | 63.6 | 52.3 | 58.4 | 65.9 | **70.0** | 52.9 | 88.2 |
| | FgFactV | 50.9 | 54.1 | **58.2** | 42.1 | 49.5 | 51.1 | 42.0 | 48.2 | 48.4 | 53.9 | 41.4 | 44.1 | 53.3 | 94.3 |
| | AnsCls | 52.7 | 57.0 | **59.6** | 48.0 | 42.3 | 53.0 | 41.4 | 46.4 | 51.4 | 48.6 | 43.2 | 44.6 | 52.9 | 88.2 |
| Llama 2 70B | MathGen | 50.0 | 49.1 | 66.9 | 20.2 | 48.3 | 54.8 | 47.8 | 64.1 | 50.8 | 74.7 | 83.4 | **85.9** | 68.0 | 97.1 |
| | FgFactV | 59.5 | 67.2 | **71.7** | 54.7 | 37.3 | 59.4 | 28.0 | 38.6 | 40.3 | 45.3 | 37.0 | 60.2 | 68.8 | 100.0 |
| | AnsCls | 33.1 | 68.1 | 50.6 | 51.7 | 29.2 | 47.5 | 20.6 | 20.0 | 24.8 | 29.5 | 54.4 | **68.3** | 69.5 | 100.0 |

Table 8: Error detection performance of LLMs with zero-shot prompts on *ReaLMistake* in **accuracy**. This table includes the additional results of the experiment in Table 4.

# G  Additional Results for Error Detection Analysis

This section includes additional results for the analysis of error detection in Section 4.

## G.1  Additional Results for Comparison to Other Tasks

Figure 9 shows the full version of Figure 5, which compares the error detection performance on *ReaLMistake* with popular evaluation benchmarks: MMLU (five-shot) (Hendrycks et al., 2021) and LMSYS Chatbot Arena Elo Rating (Zheng et al., 2023). These results consistently show that strong LLMs on MMLU and LMSYS Elo Rating detect mistakes made by LLMs at high precision but low recall (Section 4.1). Table 9 shows that precision and recall have positive and negative correlations with these metrics in both Pearson and Spearman correlation coefficients. We observe that performance on MathGen shows different trends in recall (weaker negative correlation for GPT-4 responses and positive correlation for Llama 2 70B responses) which again indicates that *ReaLMistake* includes diverse error detection tasks with different properties.

Table 10 includes LMSYS Chatbot Arena Elo Rating and MMLU performance (five-shot) of 12 LLMs used in Figure 5 and Figure 9. We accessed LMSYS Chatbot Arena on March 16, 2024.[9] Sources of MMLU performances are provided in the table.

| *ReaLMistake* | | Precision | | Recall | |
|---|---|---|---|---|---|
| Response Model | Task | Pearson | Spearman | Pearson | Spearman |
| | MathGen | 0.77* | 0.90* | 0.12 | −0.03 |
| GPT-4-0613 | FgFactV | 0.72* | 0.78* | −0.66* | −0.77* |
| | AnsCls | 0.64* | 0.58* | −0.67* | −0.75* |
| | MathGen | 0.59* | 0.69* | 0.67* | 0.48 |
| Llama 2 70B | FgFactV | 0.50 | 0.51 | −0.47 | −0.53 |
| | AnsCls | 0.63* | 0.43 | −0.20 | −0.22 |

(a) LMSYS Chatbot Arena Elo Rating vs. Error Detection Performance on *ReaLMistake*

| *ReaLMistake* | | Precision | | Recall | |
|---|---|---|---|---|---|
| Response Model | Task | Pearson | Spearman | Pearson | Spearman |
| | MathGen | 0.78* | 0.91* | −0.03 | −0.32 |
| GPT-4-0613 | FgFactV | 0.73* | 0.69* | −0.70* | −0.77* |
| | AnsCls | 0.54 | 0.47 | −0.73* | −0.80* |
| | MathGen | 0.62* | 0.77* | 0.57 | 0.28 |
| Llama 2 70B | FgFactV | 0.33 | 0.51 | −0.68* | −0.64* |
| | AnsCls | 0.48 | 0.23 | −0.52 | −0.61* |

(b) Performance on MMLU vs. Error Detection Performance on *ReaLMistake*

Table 9: Correlation between precision and recall on the error detection task in *ReaLMistake* and performance on MMLU (Hendrycks et al., 2021) or LMSYS Chatbot Arena Elo Rating (Zheng et al., 2023) on 12 LLMs. Precision and recall on *ReaLMistake* have positive and negative correlations with these metrics both in Pearson and Spearman correlation coefficients, especially in the error detection task on responses from GPT-4-0613. *: $p < 0.05$.

---

[9]https://huggingface.co/spaces/lmsys/chatbot-arena-leaderboard

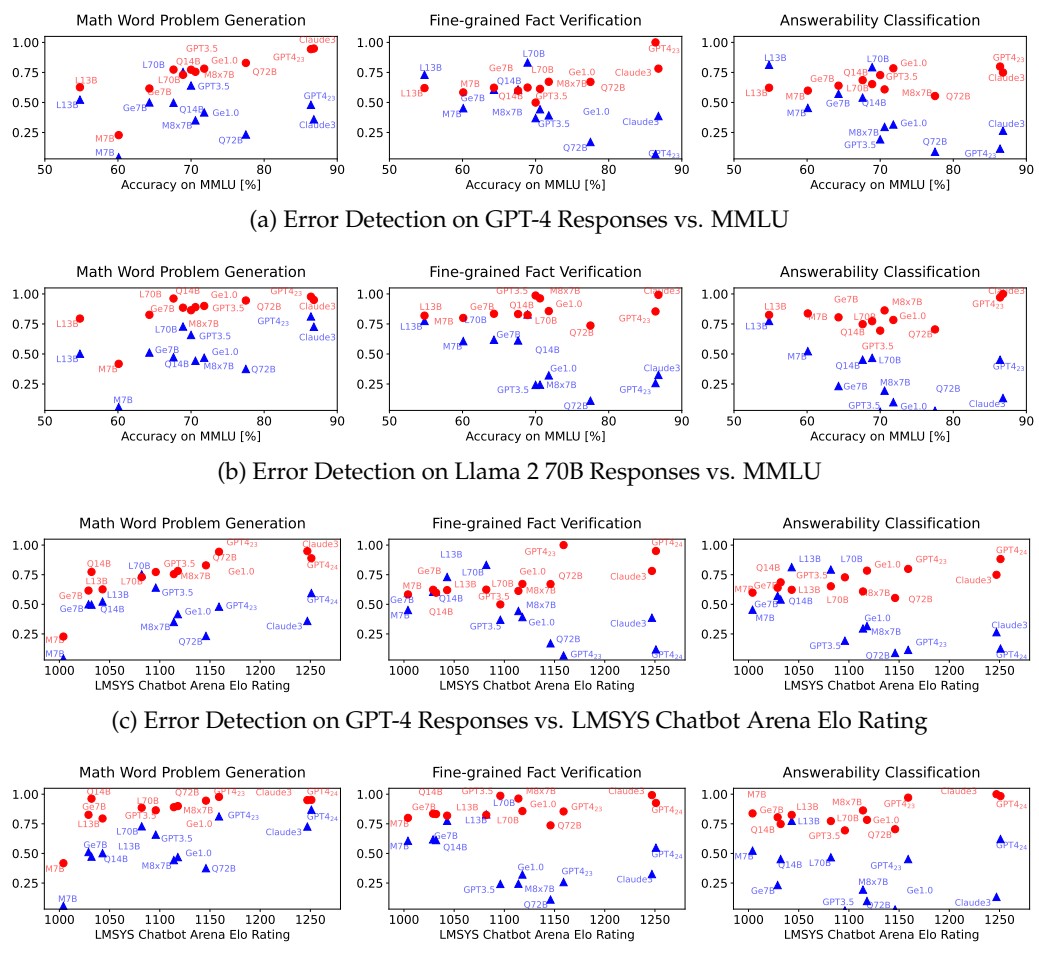

(a) Error Detection on GPT-4 Responses vs. MMLU

(b) Error Detection on Llama 2 70B Responses vs. MMLU

(c) Error Detection on GPT-4 Responses vs. LMSYS Chatbot Arena Elo Rating

(d) Error Detection on Llama 2 70B Responses vs. LMSYS Chatbot Arena Elo Rating

Figure 9: Relationship between precision and recall on the error detection task in *ReaLMistake* and performance on MMLU and LMSYS Chatbot Arena Elo Rating. Stronger LLMs on MMLU (Hendrycks et al., 2021) and LMSYS Chatbot Arena Elo Rating (Zheng et al., 2023) detect errors with higher precision but with lower recall. This is the full version of Figure 5.

| LLMs | LMSYM Elo Rating | MMLU | Source of MMLU Performance |
|---|---|---|---|
| GPT-4-0125 | 1251 | – | – |
| Claude 3 Opus | 1247 | 86.8 | (Anthropic, 2024) |
| GPT-4-0613 | 1159 | 86.4 | (OpenAI, 2023)* |
| Gemini 1.0 Pro | 1118 | 71.8 | (Gemini Team Google, 2023) |
| GPT-3.5 Turbo | 1096 | 70.0 | (OpenAI, 2023)* |
| Qwen1.5 72B | 1146 | 77.5 | (Qwen Team, 2024) |
| Mixtral 8x7B | 1114 | 70.6 | (Jiang et al., 2024) |
| Llama 2 70b | 1082 | 68.9 | (Touvron et al., 2023) |
| Llama 2 13b | 1043 | 54.8 | (Touvron et al., 2023) |
| Qwen1.5 14B | 1032 | 67.6 | (Qwen Team, 2024) |
| Gemma 7B | 1029 | 64.3 | (Google, 2024) |
| Mistral 7B | 1004 | 60.1 | (Jiang et al., 2023) |

Table 10: LMSYM Elo Rating and MMLU performance of 12 LLMs. *: The performances of previous versions of GPT-4 and GPT-3.5 in March 2023.

## G.2    Additional Results for Bias Analysis

Figure 10 and 11 show the additional results of the analysis of biases caused by small changes in prompts on the error detection task in *ReaLMistake*. These figures show that all results are consistent with the analysis in Section 4.3. As discussed in Section 4.3, these results show that recall is strongly affected by small changes in prompts (wordings and the order of binary label options) but there is no consistent trend in precision. In addition, we observe that GPT-4 (red markers) and Claude 3 (yellow star markers) are less affected by small differences in prompts.

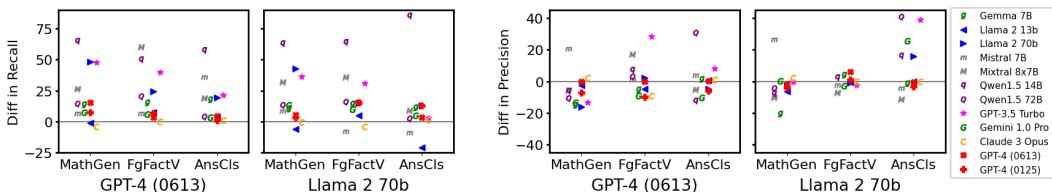

Figure 10: **Wording bias:** The values in the figures represent the recall (left) and precision (right) of prompt type 1 ("detect errors") minus type 2 ("evaluate the response"). Prompt type 2 decreases recall on most models, but there is no consistent trend in precision.

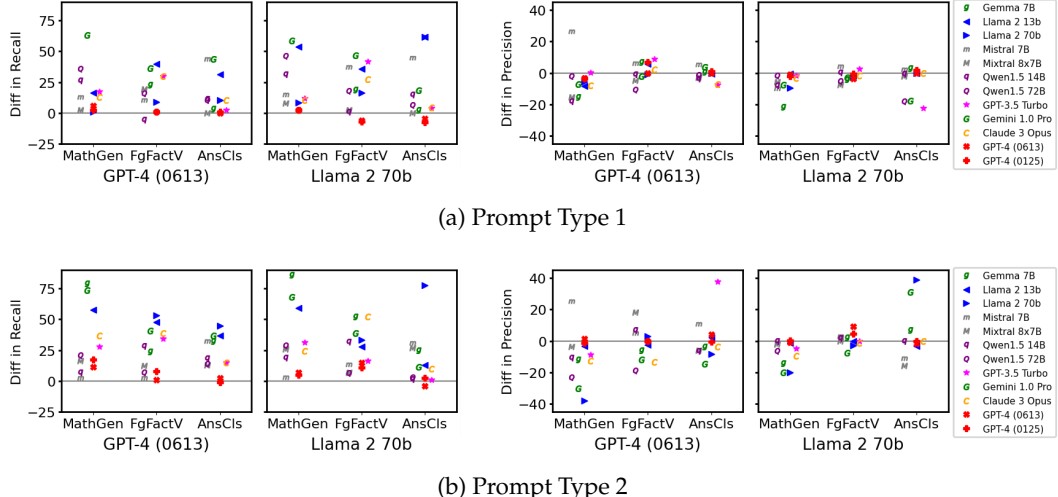

Figure 11: Performance of prompts with different option orders shows **positional bias**. The values in the figures represent the recall (left) and precision (right) of prompt type B ("no error" option first) minus prompt type A ("error" option first). In most models, putting the "no error" option before the "error" option increases recall (more errors are missed by error detection). However, there is no consistent trend in precision.

# H  Model Access Details

The model responses included in *ReaLMistake* were collected between February 1 and 21, 2024. The model responses for evaluating the error detection performance in Section 4 were collected between February 28 and March 20, 2024.

We use temperature 0.0 (`do_sample=False` for open-source models). For generating responses in *ReaLMistake* (GPT-4 and Llama 2 70B), we use the maximum possible number of new tokens for each model. For LLM-based error detection, we use the maximum number of new tokens 4,096 for all models.

## H.1  Closed Source LLMs

**OpenAI's Models.**  We experiment with GPT-3.5 Turbo (Brown et al., 2020; Ouyang et al., 2022) and GPT-4 (OpenAI, 2023) through the OpenAI's API.[10] Specifically, we use `gpt-4-0613` for generating the initial responses in *ReaLMistake*. We also use `gpt-3.5-turbo-0613` and `gpt-4-0125-preview` for error detection in Section 4.

**Claude 3 Opus.**  We experiment with Claude 3 Opus (Anthropic, 2024) through the Claude API.[11] Specifically, we use `claude-3-opus-20240229`.

**Gemini 1.0 Pro.**  We experiment with Gemini 1.0 Pro (Gemini Team Google, 2023) through Google's Vertex AI API.[12] Specifically, we use `gemini-1.0-pro`.

## H.2  Open Source LLMs

We use open-source LLMs provided on Hugging Face Hub.[13] We generate prompts by using the `apply_chat_template` function provided by the tokenizer for each model without using any system prompt. We evaluate the open-source models by using eight NVIDIA A100 SXM4 GPUs.

**Llama 2.**  We use Llama 2 70B (Touvron et al., 2023) for generating the initial responses in *ReaLMistake*. For error detection, we also evaluate Llama 2 13B. Specifically, we use `meta-llama/Llama-2-13b-chat-hf` and `meta-llama/Llama-2-70b-chat-hf`.

**Gemma 7B.**  We use Gemma 7b model (Google, 2024) for evaluating error detection. Specifically, we use `google/gemma-7b-it`.

**Mixtral 8x7B.**  We use Mistral 7B (Jiang et al., 2023) and Mixtral 8x7B (Jiang et al., 2024) for evaluating error detection. Specifically, we use `mistralai/Mistral-7B-Instruct-v0.1` and `mistralai/Mixtral-8x7B-Instruct-v0.1`.

**Qwen 1.5.**  We use Qwen 1.5 (14B and 72B) (Bai et al., 2023; Qwen Team, 2024) for evaluating error detection. Specifically, we use `Qwen/Qwen1.5-14B-Chat` and `Qwen/Qwen1.5-72B-Chat`.

---

[10]`https://platform.openai.com`

[11]`https://www.anthropic.com/api`

[12]`https://cloud.google.com/vertex-ai`

[13]`https://huggingface.co/models`

# I  Data Example of *ReaLMistake* and Error Detection Outputs

This section shows examples from *ReaLMistake* in Figure 2 and outputs from LLM-based error detectors (Section 4). The manual analysis for responses from LLM-based error detectors categorizes mistakes made by detectors as in Figure 6.

## I.1  Math Word Problem Generation (MathGen)

**MathGen - Input**

Generate a math word problem that satisfies the following requirements. First, provide the generated question. Second, generate a step-by-step solution for the generated question.

* When the requirements instruct only to use integers in the problem or solution, you may use integers with units (e.g., 15kg, 10%) but should not use fractions (e.g., 1/2) if not specified. This applies to all data types.
* The term "fractions" in the requirements refers to the numbers expressed using an integer numerator and denominator, and does not include decimals.

General Requirements:
* The generated question should not include sub-problems.
* The generated question should be valid and answerable.
* The generated question should not be ambiguous and the final answer to the generated questions must be unique.
* The generated question should not include information that is not necessary to answer the question.
* The solution should not include any approximation.
* The solution should include only one solution without any unnecessary procedures. You should not include a verification of the final answer after solving the problem.
* The question and solution should use reasonable data types and values for the domain of the question. For example, if the question is about the number of people, the answer should be a positive integer.

Specific Requirements:
* The problem requires an understanding of relative speed and time in a circular track scenario.
* The problem statement should include the phrase "Marla completes 32 laps around the track per hour".
* The solution involves division and rounding off to the nearest whole number.
* The solution should include [integers, numbers with one decimal place, numbers with 2 decimal places, fractions] and should not include any other type of number.

---

**MathGen - Response from GPT-4**

Question:
Marla completes 32 laps around the track per hour. If the track is 400 meters long, how many kilometers does Marla run in 2.5 hours?

Solution:
Step 1: First, we need to find out how many meters Marla runs in one hour. Since she completes 32 laps and each lap is 400 meters, we multiply these two numbers together.
32 laps/hour * 400 meters/lap = 12800 meters/hour

Step 2: To convert meters to kilometers, we divide by 1000 (since there are 1000 meters in a kilometer).
12800 meters/hour ÷ 1000 = 12.8 kilometers/hour

Step 3: Now, we need to find out how many kilometers Marla runs in 2.5 hours. We multiply her speed by the time.
12.8 kilometers/hour * 2.5 hours = 32 kilometers

So, Marla runs 32 kilometers in 2.5 hours.

---

**Human Expert Annotation on the Response from GPT-4**

- Error Label: error
- Error Categories: Instruction-Following
- Explanation: The question does not involve relative speed and rounding off to the nearest whole number, which are in the requirements. In addition, the solution does not include numbers with two decimal places.

---

**Error Detector based on GPT-4-0125 with Prompt Type 1-A**

The model response violates the specific requirement that the solution should include "[integers, numbers with one decimal place, numbers with 2 decimal places, fractions]" and should not include any other type of number. The solution provided uses a time value of "2.5 hours," which is a number with one decimal place, aligning with the requirements. However, the specific requirements also mention that the solution involves division and rounding off to the nearest whole number, which was not adhered to in the provided solution. The solution directly calculates the distance Marla runs in 2.5 hours without involving any rounding off to the nearest whole number as instructed.

Therefore, the model response contains an error.

Manual Analysis: ● Correct binary label, insufficient reasoning

The response does not mention why the fourth specific requirement is not satisfied. In addition, it does not mention the first requirement about "relative speed".

**Error Detector based on Claude 3 Opus with Prompt Type 1-A**

The model response follows all the general and specific requirements outlined in the model input. The generated question is valid, unambiguous, and does not include unnecessary information. The solution is step-by-step, does not involve approximations, and includes only one solution without unnecessary procedures.

The specific requirements are also met:
- The problem involves relative speed and time in a circular track scenario.
- The phrase "Marla completes 32 laps around the track per hour" is included in the problem statement.
- The solution involves division and rounding off to the nearest whole number (32 kilometers in the final answer).
- The solution includes integers (32 laps/hour, 400 meters/lap, 1000 meters/kilometer) and numbers with one decimal place (12.8 kilometers/hour), as required. It does not include any other type of number.

Therefore, the model response contains no error.

Manual Analysis: ● Wrong binary label, wrong reasoning

The detector misses some requirements (e.g., data types) and makes a mistake by saying "The problem involves relative speed".

**Error Detector based on Qwen 1.5 72B with Prompt Type 1-A**

The model response presents a math word problem involving relative speed and time on a circular track, as instructed. It includes the required phrase "Marla completes 32 laps around the track per hour" and provides a step-by-step solution that involves division and rounding to the nearest whole number. The solution uses integers, numbers with one decimal place, and numbers with two decimal places, as specified. There are no sub-problems, unnecessary information, or ambiguous elements in the question, and the answer is unique.

Therefore, the model response contains no error.

Manual Analysis: ● Wrong binary label, wrong reasoning

The detector makes a mistake by saying "The solution uses ... numbers with two decimal places".

**Error Detector based on Mixtral 8x7B with Prompt Type 1-A**

The model response contains no error. The question and solution meet all the requirements specified in the model input. The question is clear, concise, and unambiguous, and the solution is accurate and correctly follows the steps outlined. The solution uses only integers and decimal numbers as specified, and the final answer is rounded off to the nearest whole number. The problem also correctly includes the phrase "Marla completes 32 laps around the track per hour" as required. Therefore, the model response contains no error.

Manual Analysis: ● Wrong binary label, wrong reasoning

The detector makes a mistake by saying " the final answer is rounded off to the nearest whole number".

**Error Detector based on Llama 2 70B with Prompt Type 1-A**

The model response contains an error.

The error is in step 3 of the solution, where the model divides the speed by the time to find the distance traveled. The correct calculation should be:

12.8 kilometers/hour * 2.5 hours = 3200 meters

Then, to convert meters to kilometers, we divide by 1000:

3200 meters ÷ 1000 = 3.2 kilometers

Therefore, Marla runs 3.2 kilometers in 2.5 hours, not 32 kilometers as the model response states.

The specific requirements of the model input instructed the model to use integers, numbers with one decimal place, numbers with 2 decimal places, and fractions. The model response uses integers and numbers with one decimal place, but not fractions. This is not an error, but it is worth noting that the model could have used fractions to provide a more precise answer.

In conclusion, the model response contains an error in the calculation of the distance traveled and should be corrected to 3.2 kilometers.

Manual Analysis: ● Wrong binary label, wrong reasoning

The detector makes many mistakes.

---

**MathGen - Response from Llama 2 70B**

Question:
Marla competes in a bicycle race on a circular track with a diameter of 800 meters. She completes 32 laps around the track per hour. If she maintains a constant speed throughout the race, how many minutes will it take her to complete one lap?

Solution:
Let's start by converting the diameter of the track from meters to kilometers:
800 m = 0.8 km

Next, we can calculate Marla's speed in kilometers per hour:
Speed = Distance / Time
= 0.8 km / 1 h
= 0.8 km/h

Now, we need to convert Marla's speed from kilometers per hour to meters per second:
Speed (m/s) = Speed (km/h) x 1000
= 0.8 km/h x 1000
= 800 m/s

Since Marla completes 32 laps per hour, we can calculate the time it takes her to complete one lap:
Time = Distance / Speed
= 0.8 km / 800 m/s
= 0.8 s

To find out how many minutes it takes Marla to complete one lap, we can divide the time in seconds by 60:
Minutes = Time (s) / 60
= 0.8 s / 60
= 0.0133 min

Rounding up to the nearest whole number, we get:
Minutes = 1 min

Therefore, it will take Marla approximately 1 minute to complete one lap around the track.

---

**Human Expert Annotation on the Response from Llama 2 70B**

- Error Label: error
- Error Categories: Instruction-Following, Reasoning Correctness
- Explanation: The question does not involve relative speed. It also includes unnecessary information (the length of the track is not necessary). The solution makes mistakes in calculating her speed.

## I.2    Fine-grained Fact Verification (FgFactV)

**FgFactV - Input**

We provide a pair of a claim and evidence. The claim is a sentence in a Wikipedia article, and the evidence includes sentences retrieved from web articles.
Your task is to check all pieces of information in the claim and state reasoning on whether each part of the claim is supported by the evidence or not.

* Your response should end with one of the following sentences:
* "Therefore, the claim is supported." - Select this option if all information in the claim is supported by the evidence.
* "Therefore, the claim is not supported." - Select this option unless all information in the claim is supported by the evidence. In other words, select this option if any part of the claim is not mentioned in the evidence or is contradicted by the evidence.
* "Therefore, it is not clear whether the claim is supported or not." - Select this option if you cannot conclude whether the claim is supported or not because of ambiguity in the verification process.
* In your reasoning, provide line numbers of the evidence sentences supporting each part of the claim.
* You should not use any external knowledge other than the provided evidence.
* Your response should not include anything that is not necessary for verifying the claim.

Claim (a sentence in the section "Background." in the article "Mick Adams"):
Adams was born in Widnes, Lancashire, England, and he died aged 65 in Tamworth, New South Wales, Australia.

Evidence (sentences retrieved from web articles):
line 0: (meta data) TITLE: Mick Adams: Ex-Widnes rugby league captain dies, aged 65 - BBC Sport
line 2: ## Accessibility links
line 3: BBCSPORT All Sport All Sport Home Football Formula 1 Cricket Rugby U Tennis Golf Athletics Cycling BBC Sport Home Football Formula 1 Cricket Rugby Union Tennis Golf Athletics Cycling
line 6: ### Events
line 7: Winter Olympics Commonwealth Games Men's World Cup Women's World Cup
line 9: England Scotland Wales Northern Ireland
line 10: ### World Sport
line 11: European Football Champions League African Football Sport Africa Rugby League Scores & Fixtures Tables All Teams Scores & Fixtures Tables All Teams Rugby League Home
line 12: # Mick Adams: Ex-Widnes rugby league captain dies, aged 65
line 14: Mick Adams was a Great Britain international, and former captain at Widnes
line 17: He was part of four Wembley triumphs, and scored 68 tries in more than 400 appearances - culminating in 1992's addition to the Chemics' Hall of Fame.
line 18: The back-rower, who passed away on Thursday, also represented Great Britain and England during his career.
line 19: "The club would like to pass on their sincere condolences to Mick's family and friends at this time," a statement said.
line 20: Widnes will hold a minute's applause for Adams before Sunday's Super League fixture against Castleford.
line 21: Former Widnes winger Martin Offiah paid his respect to Mick Adams
line 22: Former Widnes team-mate Eddie Cunningham gave this tribute to Adams
line 23: ## Find out more
line 24: ### The Super League Show

---

**FactV - Response from GPT-4**

The claim states that Mick Adams was born in Widnes, Lancashire, England, and died aged 65 in Tamworth, New South Wales, Australia. The evidence supports the first part of the claim, confirming that Adams was indeed born in Widnes, England (line 0, line 14). However, the evidence does not provide any information about where Adams died, specifically whether it was in Tamworth, New South Wales, Australia. Therefore, it is not clear whether the claim is supported or not.

---

**Human Expert Annotation on the Response from GPT-4**

- Error Label: error
- Error Categories: Instruction-Following, Reasoning Correctness
- Explanation: Line 14 does not mention that he was born in Widnes. The response does not mention his age when he died. In addition, the conclusion should be "not supported".

---

**Error Detector based on GPT-4-0125 with Prompt Type 1-A**

The model response correctly identifies that the evidence supports the claim that Mick Adams was born in Widnes, England, as indicated by lines 0 and 14. However, the model response incorrectly states that the evidence does not provide any information about where Adams died. The evidence does not explicitly mention Tamworth, New South Wales, Australia, as the place of death, which is a crucial part of the claim that needs verification. Since the model response concludes that "it is not clear whether the claim is supported or not" without verifying the specific location of Adams' death, it fails to follow the instruction that requires a clear conclusion based on the evidence provided. The correct conclusion, based on the instructions and the evidence provided, should have been that the claim is not supported, as there is no evidence to confirm the part of the claim regarding the place of death.

Therefore, the model response contains an error.

Manual Analysis: ● Correct binary label, wrong reasoning

The detector makes a mistake by saying "the evidence supports the claim that Mick Adams was born in Widnes, England" and "the model response incorrectly states that the evidence does not provide any information about where Adams died".

---

**Error Detector based on Claude 3 Opus with Prompt Type 1-A**

The model response correctly identifies that the evidence supports Adams being born in Widnes, England. It also correctly points out that the evidence does not mention where Adams died, and therefore it cannot be determined if the claim about his death location is supported or not.

However, the model response concludes with "Therefore, it is not clear whether the claim is supported or not." This does not follow the instructions in the model input, which state that the response should end with either "Therefore, the claim is supported.", "Therefore, the claim is not supported.", or "Therefore, it is not clear whether the claim is supported or not." The model should have concluded with "Therefore, the claim is not supported." since part of the claim (the death location) is not supported by the given evidence.

Therefore, the model response contains an error.

Manual Analysis: ● Correct binary label, wrong reasoning

The detector makes a mistake by saying "The model response correctly identifies that the evidence supports Adams being born in Widnes, England".

**Error Detector based on Qwen 1.5 72B with Prompt Type 1-A**

The model response correctly identifies that the evidence supports the fact that Mick Adams was born in Widnes, England. However, it also correctly acknowledges that there is no information in the evidence about where Adams died. The model response appropriately concludes with "Therefore, it is not clear whether the claim is supported or not." This indicates that the model has followed the instructions and requirements in the model input. Therefore, the model response contains no error.

Manual Analysis: ● Wrong binary label, wrong reasoning

The detector makes a mistake by saying "The model response correctly identifies that the evidence supports the fact that Mick Adams was born in Widnes" and "The model response appropriately concludes with "Therefore, it is not clear whether the claim is supported or not."".

**Error Detector based on Mixtral 7x8B with Prompt Type 1-A**

The model response correctly identifies that the first part of the claim is supported by the evidence. However, the response should not conclude that it is not clear whether the claim is supported or not just because the evidence does not mention where Adams died. The model should only make a determination about the supported status of the claim based on the provided evidence. Since the evidence does not contradict the claim that Adams died in Tamworth, New South Wales, Australia, it is possible that the claim is true even if the evidence does not specifically support it. Therefore, the model response contains an error.

Manual Analysis: ● Correct binary label, wrong reasoning

The detector makes a mistake by saying "The model response correctly identifies that the first part of the claim is supported". In addition, reasoning of "Since the evidence does not contradict the claim..." is wrong.

**Error Detector based on Llama 2 70B with Prompt Type 1-A**

The model response contains an error.

The error in the model response is the conclusion that it is not clear whether the claim is supported or not. The evidence provided in the model input does not mention anything about Mick Adams' death or where he died. Therefore, the model response should have concluded that the claim is not supported, as there is no evidence to support the second part of the claim regarding Adams' death.

The correct response should be:

"Therefore, the claim is not supported."

The evidence only supports the first part of the claim, which states that Mick Adams was born in Widnes, Lancashire, England. The evidence does not provide any information about Adams' death or where he died, therefore, it does not support the second part of the claim.

Manual Analysis: ● Correct binary label, wrong reasoning

The detector makes a mistake by saying "The evidence only supports the first part of the claim".

---

**FgFactV - Response from Llama 2 70B**

Therefore, it is not clear whether the claim is supported or not.

Reasoning:

* The first part of the claim, "Adams was born in Widnes, Lancashire, England," is not directly supported by the provided evidence. None of the evidence sentences mention Adams' birthplace or date of birth.
* The second part of the claim, "and he died aged 65 in Tamworth, New South Wales, Australia," is also not directly supported by the provided evidence. While the evidence does mention Adams' death age (line 18) and location (line 19), it does not explicitly state his age at the time of death as 65.
* However, some evidence sentences do support the idea that Adams was a well-known rugby player, which could indirectly support the claim that he was born and died in the places mentioned. For example, line 14 mentions Adams' role as a former captain at Widnes, and line 17 highlights his achievements in the sport.

Overall, while there is no direct evidence supporting every part of the claim, there is enough related information in the evidence to suggest that the claim might be true. Yet, due to the lack of explicit support, it is still unclear whether the claim is entirely accurate. Hence, the response is "Therefore, it is not clear whether the claim is supported or not."

---

**Human Expert Annotation on the Response from Llama 2 70B**

- Error Label: error
- Error Categories: Instruction-Following, Reasoning Correctness
- Explanation: The conclusion should be "not supported". In addition, the evidence does not include Eric's name. In addition, it misses hid death age in line 1. Line 19 does not support the death place.

I.3    Answerability Classification (AnsCls)

**AnsCls - Input**

Answer the following question. Assume you are on Jan 18, 2018 and questions that require knowledge after this date should be classified as unanswerable. Provide reasoning first, then state your answer in the end.
* Your reasoning should not include any factual errors.
* If the question includes incorrect information or is invalid (e.g., inconsistent date assumption or ambiguous due to insufficient information), your answer at the end of your response should clearly state that the question is unanswerable.
* If a future event is asked in the past tense, the question should be classified as unanswerable.

Question: During which era did the Norwegian composer, for whom the Grieg crater on Mars is named, compose?

**AnsCls - Response from GPT-4**

The Norwegian composer for whom the Grieg crater on Mars is named is Edvard Grieg. Edvard Grieg was a composer during the Romantic era of music, which spanned from approximately 1800 to 1900. Grieg was born in 1843 and died in 1907, so his active years as a composer fell within this period. Therefore, the era during which Grieg composed was the Romantic era.

Answer: The Romantic era.

**Human Expert Annotation on the Response from GPT-4**

- Error Label: error
- Error Categories: Parameterized Knowledge
- Explanation: The crater is on Mercury.

**Error Detector based on GPT-4-0125 with Prompt Type 1-A**

The model response accurately identifies Edvard Grieg as the Norwegian composer for whom the Grieg crater on Mars is named and correctly states that he composed during the Romantic era, which is consistent with the historical period of approximately 1800 to 1900. The response provides a clear reasoning for the answer, mentioning Grieg's lifespan (1843-1907) and situating his active years within the Romantic era. The question did not include incorrect information, was not invalid, and did not ask about a future event in the past tense. Therefore, the model response follows all instructions and requirements in the model input.

Therefore, the model response contains no error.

Manual Analysis:  ●  Wrong binary label, wrong reasoning

The detector makes a mistake by saying "The model response accurately identifies Edvard Grieg as the Norwegian composer for whom the Grieg crater on Mars is named ...".

**Error Detector based on Claude 3 Opus with Prompt Type 1-A**

The model response correctly identifies Edvard Grieg as the Norwegian composer for whom the Grieg crater on Mars is named. It also accurately states that Grieg composed during the Romantic era, which spanned from approximately 1800 to 1900, and that Grieg's life (1843-1907) fell within this period. The response provides clear reasoning and a concise answer to the question, without including any factual errors or inconsistencies with the given date assumption of January 18, 2018.

Therefore, the model response contains no error.

Manual Analysis: ● Wrong binary label, wrong reasoning

The detector makes a mistake by saying "The model response correctly identifies Edvard Grieg as the Norwegian composer for whom the Grieg crater on Mars is named".

**Error Detector based on Qwen 1.5 72B with Prompt Type 1-A**

The Norwegian composer Edvard Grieg, to whom the Grieg crater on Mars is dedicated, indeed lived and composed during the Romantic era, which lasted from around 1800 to 1900. Grieg's birth year was 1843, and he passed away in 1907, so his compositional career fell within this time frame. The model response accurately provides the reasoning and the answer. Therefore, the model response contains no error.

Manual Analysis: ● Wrong binary label, wrong reasoning

The detector makes a mistake by saying "The Norwegian composer Edvard Grieg, to whom the Grieg crater on Mars ...".

**Error Detector based on Mixtral 7x8B with Prompt Type 1-A**

The model response provides accurate information about Edvard Grieg and the Romantic era of music. However, the model input states that the answer should be provided assuming the date is January 18, 2018. The response does not explicitly state that the answer is based on this assumption. Although the answer is correct, the lack of explicit reference to the date assumption is a minor issue that could potentially lead to confusion.

Therefore, the model response contains an error.

Manual Analysis: ● Correct binary label, wrong reasoning

It is true that the model response does not provide explicit reasoning on the date assumption, but it does not mean that the response is wrong.

### Error Detector based on Llama 2 70B with Prompt Type 1-A

The model response contains an error.

The error in the model response lies in the fact that it assumes that the question is asking about the composer's active years as a composer, which is not explicitly stated in the question. The question only asks when the composer composed, without specifying whether it refers to their active years or not. Therefore, the model response's assumption that the composer's active years fall within the Romantic era is not necessarily true.

Additionally, the model response does not address the fact that the question is asking about a future event (the composition of music) in the past tense. This inconsistency in tense should be addressed in the response, and the model should recognize that the question is unanswerable due to this ambiguity.

Therefore, the model response contains an error, and it should be classified as such.

Manual Analysis: ● Correct binary label, wrong reasoning

There are many mistakes.

### AnsCls - Response from Llama 2 70B

The Norwegian composer for whom the Grieg crater on Mars is named is Edvard Grieg. He was born in 1843 and died in 1907. Therefore, he composed during the Romantic era, specifically from the mid-19th century to the early 20th century.

However, it's important to note that the Grieg crater on Mars was not named until much later, in 1973, by the International Astronomical Union. So while Edvard Grieg did compose music during his lifetime, it's not possible for him to have composed music during the era when the crater was named, as that would have required him to still be alive and composing decades after his death.

Therefore, I must classify this question as unanswerable due to its reliance on an impossible scenario.

### Human Expert Annotation on the Response from Llama 2 70B

- Error Label: error
- Error Categories: Reasoning Correctness
- Explanation: The reasoning in the second paragraph of the response is invalid.

