# OpenReview forum: "Evaluating LLMs at Detecting Errors in LLM Responses"
_colmweb.org/COLM/2024/Conference — COLM_

### Official Review · Reviewer_BGkA · 2024-05-10

**Rating:** 6
**Confidence:** 3
**Ethics Flag:** 1

**Summary:**

In their paper, the authors introduce an error detection benchmark called ReaLMistake. This benchmark evaluates four distinct types of errors—reasoning correctness, instruction-following, context-faithfulness, and parameterized knowledge—made by large language models (LLMs) across three selective natural language processing (NLP) tasks. The results reveal several key findings:

1) Top-performing LLMs, including GPT-4, exhibit low recall in error detection tasks.
2) While LLMs can correctly identify errors, their explanations remain unreliable.
3) The performance of error detection is notably sensitive to subtle variations in prompts.
4) Previous methods aimed at enhancing error detectors have proven ineffective.

**Questions To Authors:**

1) In Table 2, I notice that the dataset is biased, especially the error percentage of the Llama2 response is super high (75.0, 78.8, 77.5). How different data distribution affects the error detection results? What is the reasonable range of the error percentage?

2) A following question of 1)： In Table 4, is it suitable to use your defined "Random" recall score as the baseline? Your dataset with Llama 2 is super biased and to give a more extreme example, if the error percentage is 100%, no model can beat the recall baseline(100). However, many models work better than random guess (recall=50). In this case, I think your claim "LLM-based error detectors often perform worse than random baselines" is a bit unsound.

**Reasons To Accept:**

1） The main contribution of this paper is to introduce a novel benchmark, ReaLMistake, for error detection in LLM responses. The benchmark contains various, objective and realistic errors made by LLMs, which is valuable for future research in this area.
2) It provides an extensive analysis of the performances of 11 different LLMs, offering valuable insights into the error detection capabilities of popular LLMs.
3) The paper is well-organized, presenting sufficient experiments, ablations, and detailed analysis.

**Reasons To Reject:**

The LLMs employed to generate the response (GPT4 and Llama2) could impact the reliability of the benchmark. Another concern is about the sample size. The ReaLMistake benchmark contains a relatively small number of samples compared to other benchmarks with binary labels (e.g., SummEdits and BIG-Bench Mistake).

---

> ### Author Rebuttal · Authors · 2024-05-30
>
> We appreciate your insightful comments and valuable feedback!
>
> > The LLMs employed to generate the response (GPT4 and Llama2) could impact the reliability of the benchmark.
>
> I would appreciate it if you could elaborate on this point. We select GPT-4 and Llama 2 70B because they are widely adopted, and the study of error detection on their responses is well-motivated.
>
> > The ReaLMistake benchmark contains a relatively small number of samples compared to other benchmarks with binary labels (e.g., SummEdits and BIG-Bench Mistake).
>
> While we agree that larger is better, we consider the size of our benchmark to be reasonable for the following reasons:
>
> (1) Unambiguity: ReaLMistake includes unambiguous errors with high annotation quality. We expect that a relatively small benchmark can reliably evaluate error detectors.
>
> (2) Diversity: Our benchmark includes diverse instances in three tasks and four error categories.
>
> (3) Annotation Cost: SummEdits and Big-Bench Mistake use synthetic data or relatively simple tasks, which make human annotation easier. Our work focuses on natural mistakes by LLMs, whose error annotation is much more difficult.
>
> > In Table 2, I notice that the dataset is biased [...]
>
> Our benchmark includes errors in LLM responses in the natural distribution. We consider that the label distribution is not an issue if a sufficient number of instances are included for each label. When a balanced dataset is required, we can sub-sample instances with the majority label or up-sample instances with the minority label.
>
> Our experiments suggest that our benchmark includes a reasonable number of instances for each label because (1) our experiments capture performance differences between different models and (2) the state-of-the-art models are still far from human performance.
>
> > In Table 4, is it suitable to use your defined "Random" recall score as the baseline? [...]
>
> We select our random baseline to simulate a situation in which we know the estimated error frequency of LLM responses. For example, when providers of LLM-based services try to detect mistakes in LLM responses, they often have the estimated error frequency of their LLMs and use it as a baseline.
>
> We also note that there is no universally reasonable random baseline. For example, a random baseline with 50% error probability is relatively strong for error detection on GPT-4 responses (with around 50% errors) but very weak for Llama 2 70B responses (with around 75% errors).

---

> > ### Comment · Reviewer_BGkA · 2024-06-06
> >
> > Thank you for answering my questions. I'll keep the score.

---

### Official Review · Reviewer_sN45 · 2024-05-11

**Rating:** 9
**Confidence:** 5
**Ethics Flag:** 1

**Summary:**

This paper proposes. new benchmark, ReaLMistake, that evaluates how good LLMs are at detecting objective errors. The authors first constructed three sets of instructions (math word problem generation, fine-grained fact verification, and answerability classification) and prompted GPT-4/Llama-2-70B to generate responses. Then the authors manually annotated the errors in the model responses with rationales. The error types are categorized into (1) instruction following, (2) reasoning correctness, (3) context-faithfulness, and (4) parameterized knowledge. All four are common errors of LLMs and can be objectively evaluated. The human agreement rate is high (95.9 F1).

Then the authors evaluated a number of open-source/closed-source LLMs on the benchmark -- this includes both the accuracy on detecting the errors and a manual evaluation on the rationales generated by the models. The experiment results are quite interesting: (1) even the best models struggle at accurately predicting the errors. More surprisingly, there seems to be a negative correlation between model capacity (measured by Chatbot arena score) and the error detection recall. (2) The rationales generated by models can be very wrong even if the prediction is correct, especially for open-source models. (3) Prompt engineering is very difficult for error detection and even more advanced prompting techniques like self-consistency does not help.

**Questions To Authors:**

If I understand correctly, the annotation includes the types of the error, but the performance on Table 4 and the prompts used by models are simply binary error detection. Would it make the model perform better if they are prompted with the error type definitions and are prompted to provide the error type as well? The experiments would be more insightful as well if a per-type analysis is provided.

**Reasons To Accept:**

(1) This is a well-motivated and high-quality dataset that will provide high utility to the language model community. The dataset annotations are manually curated by the authors and have a high human agreement rate. The schemas (source datasets and error types) are intuitive and comprehensive. Not only is error detection an important applications of LLMs, but this dataset (ReaLMistake) can serve as a benchmark for evaluating general LM abilities as well.

(2) The experiments are comprehensive and insightful, especially (a) the negative correlation between model capabilities and error detection recall, and (b) the prompt sensitively of the task. Our community will benefit from those insights for better understanding LLM capabilities and building better LLMs/error detection models.

**Reasons To Reject:**

(1) The source tasks are limited to synthetic tasks like question generation and fact verification. It would be more interesting to see such a dataset constructed on more diverse instructions such as ShareGPT/Alpaca/OpenAssistant.

(2) The negative correlation between the LLM capacity and error detection recall, though interesting, is a bit unintuitive. The authors also did not offer a reasonable guess of why this happens -- which makes me wonder if this is due to suboptimal designs of the prompt.

---

> ### Author Rebuttal · Authors · 2024-05-30
>
> We appreciate your insightful comments and valuable feedback!
>
> > (1) The source tasks are limited to synthetic tasks like question generation and fact verification. It would be more interesting to see such a dataset constructed on more diverse instructions such as ShareGPT/Alpaca/OpenAssistant.
>
> We design realistic and meaningful tasks in ReaLMistake to collect unambiguous errors in LLM responses, which are useful for evaluating error detectors.
>
> We agree that creating error detection benchmarks with more diverse instructions is desirable. However, collecting unambiguous error annotations in general tasks is challenging, as they are often open-ended and subjective.
>
> > (2) The negative correlation between the LLM capacity and error detection recall, though interesting, is a bit unintuitive. The authors also did not offer a reasonable guess of why this happens -- which makes me wonder if this is due to suboptimal designs of the prompt.
>
> We consider there is no easy way to improve the prompts since strong LLMs consistently show low recall on multiple prompt designs (Sections 4.3, 4.4). Here are possible explanations for the negative correlation.
>
> * Strong LLMs can correctly classify responses without mistakes as "no error", resulting in high precision. However, they are conservative about detecting mistakes, resulting in low recall.
> * Weak LLMs do not have strong error detection capabilities and may return almost random predictions for difficult cases. As a result, recall can become higher than the conservative strong LLMs, although it results in low precision.
>
> We will improve the explanation in future versions of our paper.
>
> > Would it make the model perform better if they are prompted with the error type definitions and are prompted to provide the error type as well?
>
> ReaLMistake is designed to evaluate general-purpose error detectors, which do not assume any specific target task domains or error types. This setting is motivated by a real-world situation in which providers of LLM-based services try to detect any type of mistake in LLM responses automatically.
>
> In the situation of targeting specific tasks, we agree that error detectors would be improved when designed for specific error types (e.g., prompted with the error type definitions, as you suggested).
>
> > The experiments would be more insightful as well if a per-type analysis is provided.
>
> Thank you for your suggestion. We will provide a more detailed per-type analysis in future versions.

---

> > ### Comment · Reviewer_sN45 · 2024-06-05
> > **Ack**
> >
> > Thanks for answering my questions!

---

### Official Review · Reviewer_3EKz · 2024-05-12

**Rating:** 7
**Confidence:** 4
**Ethics Flag:** 1

**Summary:**

The paper introduces a benchmark for detecting errors in LLM-generated text. The dataset is generated by modifying several existing datasets (e.g. AQuA, HotpotQA), by prompting an LLM to modify existing questions according to certain instructions. The LLM-generated responses are manually filtered for those that failed to correctly follow the instructions. After constructing this dataset, the paper evaluates a number of LLMs, and finds low recall in identifying errors. The paper also finds that, even when errors are correctly identified, the LLM-generated explanations of these errors are frequently incorrect.

**Reasons To Accept:**

- The paper is addressing an important use case. It would be useful to have LLM-based error detection in many applications.

- The method for constructing the errors is scalable and reliable. The errors are manually verified, and there is very high inter-rater agreement.

- The experimental results are interesting and convincing. The paper evaluates an appropriate set of models, and finds strikingly poor performance in most cases.

- The paper is clearly written.

**Reasons To Reject:**

- It would be interesting to understand what factors make an error hard to identify. The results in Table 4 are fairly coarse grained. Does the dataset support more fine-grained evaluations?

- This is not very important, but it would be useful to evaluate prompting strategies more comprehensively, beyond what is described in Section 4.3. It seems like it should be possible to improve recall at the cost of lower precision. How large is that cost?

- The errors are generated from a limited number of templates on three tasks. It is uncertain how generalizable the results will be to other types of errors, and those that occur in the wild. The paper is nonetheless a good starting point.

---

> ### Author Rebuttal · Authors · 2024-05-30
>
> We sincerely appreciate your insightful comments and valuable feedback!
>
> > It would be interesting to understand what factors make an error hard to identify. The results in Table 4 are fairly coarse grained. Does the dataset support more fine-grained evaluations?
>
> Our analysis shows that error detection on MathGen (which includes errors in instruction following) is relatively easy for strong models, while the other tasks are difficult for all models. It suggests that instruction-following errors are relatively easy for detectors, but other types of errors (especially context-faithfulness and knowledge) are challenging.
>
> ReaLMistake includes information that can be used for more detailed analysis: error categories and human explanations about the errors. We will provide more analysis in future versions.
>
> > [...] it would be useful to evaluate prompting strategies more comprehensively, beyond what is described in Section 4.3.
>
> Beyond Section 4.3, in Section 4.4 and Appendix A.3, we also evaluate a prompting strategy that provides detailed instructions for evaluation steps in prompts, which is motivated by G-Eval. However, it does not improve performance.
>
> We agree that more analysis of prompting strategies, such as decomposing responses, is a promising direction for future research.
>
> > It seems like it should be possible to improve recall at the cost of lower precision. How large is that cost?
>
> As shown in Figures 9 and 10, we observe that the recall of strong models (e.g., GPT-4) is not sensitive to prompt engineering. Therefore, as mentioned in Section 4.3, we conclude that there is no easy way to improve the recall of strong models, even at the cost of lower precision.
>
> > The errors are generated from a limited number of templates on three tasks. It is uncertain how generalizable the results will be to other types of errors, and those that occur in the wild. The paper is nonetheless a good starting point.
>
> ReaLMistake is suitable for evaluating error detectors because it includes unambiguous errors in LLM responses, collected using well-designed tasks. In addition, to enable our benchmark to provide generalizable analysis, we include three diverse tasks and four error categories widely observed in LLM responses.
>
> We agree that creating more diverse benchmarks is a promising direction for future research. However, collecting unambiguous errors is difficult in tasks in the wild, which are often open-ended and subjective.

---

> > ### Comment · Reviewer_3EKz · 2024-06-05
> >
> > Thank you for your response.
> >
> > >We agree that creating more diverse benchmarks is a promising direction for future research. However, collecting unambiguous errors is difficult in tasks in the wild, which are often open-ended and subjective.
> >
> > I am sympathetic to the difficulty of collecting errors in the wild, and agree that this is a good place to start. Nonetheless, it is not clear how well the results will generalize.

---

### Official Review · Reviewer_9NZk · 2024-05-15

**Rating:** 6
**Confidence:** 4
**Ethics Flag:** 1

**Summary:**

The paper introduces ReaLMistake, which the authors claim is the first error detection benchmarking suite consisting of objective, realistic and diverse errors made by LLMs. ReaLMistake contains three tasks that introduce objectively assessable errors in four categories (reasoning correctness, instruction-following, context-faithfulness, and parameterised knowledge), eliciting naturally observed and diverse errors in responses of GPT-3 and Llama 270B annotated by experts on three tasks three tasks (Math Word Problem Generation, Fine-grained Fact Verification, and Answerability Classification). The benchmark includes error annotations (binary error label, error categories, and human explanation about errors) on responses from GPT-4 and Llama 2 70B on the three tasks. The authors show that top LLMs like GPT-4 detect errors made LLMs at very low recall, and all LLM-based error detectors perform much worse than humans, that explanations by LLM-based error detectors lack reliability and LLM-based error detection is sensitive to small changes in prompts but remains challenging to improve, and finally, popular approaches to improve LLMs, including self-consistency and majority vote, do not improve the error detection performances. The authors promise to make the code and data available on acceptance.

Overall the quality of the paper is quite high in terms of the detail of the description of the dataset, though this is somewhat in part due to the lengthy Appendices making very clear what the prompts and responses were. ReaLMistake undoubtedly could be part of a good resource for evaluation for LLM error detection, though it is limited in scope, as partially admitted to by the authors. The presentation overall is quite strong and clear, though see below for some queries about it which should be addressed. It seems more incremental research than highly novel, as the authors define the benchmarking dataset as an intersection of what has gone before, however the paper claims ReaLMistake is novel in focusing on errors that are unambiguously judged as such by human annotators (though see concerns on this claim below). The approach is quite principled scientifically, though I would dispute the claim of unambiguously annotated error data as being essential for good error detection benchmarking for LLMs, and perhaps if accepted, the authors can reduce the strength of that claim and describe alternatives where multiple annotator’s judgments are included (see below).

**Questions To Authors:**

I would reduce the strength of the claims around ambiguity and frame your dataset as collecting error judgements which are low in ambiguity (as illustrated by the 95% F1 score of agreement), rather than ‘objective’. This is scientifically too strong.

The claim of it being a diverse set of errors might also be a bit strong, as it is dependent on the diversity and difficulty of the original tasks – I would reduce this and discuss that there is more work to be done to enlarge the scope of error type.

As per the above, it is not clear how novel the approach is, given Table 1’s column ticks – please further clarify how it is going further than the top row’s system in the text, particularly if you lessen the claim that human inter-subjective disagreement is a bad thing for annotation overall.

Please calculate and report Cohen’s kappa for the agreement score, which is similar to F1 but more comparable to other inter-annotator agreement scores.

There are some errors in phrasing throughout which require a proof-read - e.g.
-	Use ‘LLM-based’ (rather than LLMs-based)
-	Use determiners in places like ‘We introduce ReaLMistake benchmark’ –> ‘We introduce the ReaLMistake benchmark’

**Reasons To Accept:**

There is a useful data contribution of the 900 instances of error annotations by experts, including human explanations for the judgment of an error on responses from GPT-4-0613. This is potentially very useful for certain kinds of error detection benchmarking and is challenging, as shown by the authors who show that near state-of-the-art models such as GPT-4x exhibit a very high error rate in detecting errors in certain tasks with the prompts designed by the authors.

The statistics of the ReaLMistake benchmark are clear and show a good range of errors in the different dimensions defined by the authors.

The Appendices are very useful as a clear exemplification of prompts and further results, which when included, mean the reproducibility of the paper is quite high from the text alone.

Overall the paper is well-written, and fairly easy for readers to understand the method and approach employed.

**Reasons To Reject:**

My main reservation about the paper, which could be corrected if claims were slightly toned down, is in terms of the motivation of the form of the benchmark: the authors claim they are against human subjective ratings of errors in a response, or at least instances which are not universally agreed upon for their label (e.g. “We cannot evaluate error detectors if humans cannot determine whether LLM responses include errors”). However, this claim is too strong, and it is not clear how the inclusion of multiple (potentially differing) annotations would make things worse for a benchmark – they are certainly a challenge to be solved going forward, but rigorous inter-annotator agreement studies in creating these benchmark datasets could be a way to deal with ambiguous/non-unanimous labelling - in fact it could even be a good idea to include the differing annotations in the benchmark. The challenge would then shift away from limiting a dataset to unambiguous error instances only to developing systems that give similar confidence levels to the level of inter-annotator agreement. Even in some cases there are mathematics-based prompts, there is still room for ambiguity in the interpretation of verbal prompts.

It is not clear whether the explanations were agreed upon between the annotators.

It is not made clear how novel the benchmark is as Table 1 suggests ‘MT-Bench, PandaLM, LLMEvalˆ2 Multiple NLP Tasks’ has more criteria and in multiple domains.

---

> ### Author Rebuttal · Authors · 2024-05-30
>
> We appreciate your detailed comments and valuable suggestions!
>
> > My main reservation about the paper, which could be corrected if claims were slightly toned down [...]
>
> Thank you for your suggestion. We agree that it is technically possible to study ambiguous errors by annotating each instance with multiple annotators, which can be a future work. We will discuss this point and tone down our claim.
>
> In this work, we choose to use unambiguous errors because of the following advantages:
>
> (1) Not affected by annotators' subjectivity.
>
> (2) Reliable evaluation: Unambiguous errors can reliably evaluate error detectors because of the small noise.
>
> (3) Binary labels: We can collect binary labels, making the evaluation of binary predictions easy.
>
> (4) Annotation cost: We do not need to collect many annotations for each instance.
>
> > It is not clear whether the explanations were agreed upon between the annotators.
>
> Thank you for your suggestion. We agree that this is a promising direction for further verifying that the errors in ReaLMistake are unambiguous.
>
> > [...] Table 1 suggests ‘MT-Bench, PandaLM, LLMEvalˆ2 Multiple NLP Tasks’ has more criteria and in multiple domains
>
> > As per the above, it is not clear how novel the approach is [...]
>
> We would like to clarify that those benchmarks are for evaluating **pairwise (ranking)** evaluators, which compare multiple responses, while our benchmark is for evaluating **pointwise** evaluators. Pairwise benchmarks cannot be directly used to evaluate error detectors because they include relative evaluation between multiple responses.
>
> This table intends to show that benchmarks for **pointwise** evaluation are lacking, although there are diverse **pairwise** benchmarks. We will improve it in future versions.
>
> > I would reduce the strength of the claims around ambiguity [...]
>
> Thank you for these suggestions. We will definitely improve the use of the terms to describe the properties of our benchmark more precisely.
>
> > Please calculate and report Cohen’s kappa [...]
>
> We report F1 because we evaluate human performance, not inter-annotator agreement. Each instance in ReaLMistake is annotated by one annotator and verified by another, and after the benchmark is finalized, human performance is evaluated using error classifications by other annotators.
>
> For reference, when we consider the labels in ReaLMistake and human classification (for human performance) as two annotators, Cohen's kappa is 0.80 for GPT-4 and 0.96 for Llama 2 70B.

---

> > ### Comment · Reviewer_9NZk · 2024-06-07
> > **Response to the rebuttal**
> >
> > This is good the Cohen's kappa figures have been added and would be added to an accepted version.
> >
> > The clarification on pair-wise benchmarks vs pointwise are welcome, and again would be good to include to discuss the limitations.
> >
> > My reservation is that the claims must be toned down about the unambiguous nature of the data - no natural language data is fully unambiguous.

---

### Decision · Program_Chairs · 2024-07-10

**Decision:**

Accept

**Comment:**

This paper contributes to the timly problem of evaluating LLMs when used as evaluators. The authors introduce a welcome new error detection benchmark dataset, ReaLMistake, comprising organic errors with granular annotations. Using this, they then provide new empirical insights into the strengths and weaknesses of LLMs when used as evaluators.

Overall, this is a nice, timely contribution which will be of broad interest and use to the COLM community.